# FutureMind: Equipping Small Language Models with Strategic Thinking-Pattern Priors via Adaptive Knowledge Distillation

**Shaoxiong Yang**[1], **Junting Li**[2*], **Mengyuan Zhang**[1], **Chao Li**[1†], **Wei Liu**[1], **Jian Luan**[1]

[1]MiLM Plus, Xiaomi Inc.
[2]Department of Computer Science, Imperial College London
`{yangshaoxiong,zhangmengyuan7,lichao75,liuwei40,luanjian}@xiaomi.com`
`jl523@ic.ac.uk`

## Abstract

Small Language Models (SLMs) are attractive for cost-sensitive and resource-limited settings due to their efficient, low-latency inference. However, they often struggle with complex, knowledge-intensive tasks that require structured reasoning and effective retrieval. To address these limitations, we propose FutureMind, a modular reasoning framework that equips SLMs with strategic thinking-pattern priors via adaptive knowledge distillation from large language models (LLMs). FutureMind introduces a dynamic reasoning pipeline composed of four key modules: Problem Analysis, Logical Reasoning, Strategy Planning, and Retrieval Guidance. This pipeline is augmented by three distinct retrieval paradigms that decompose complex queries into tractable subproblems, ensuring efficient and accurate retrieval execution. Extensive experiments on multi-hop QA benchmarks, including 2WikiMultihopQA, MuSiQue, Bamboogle, and Frames, demonstrate the superiority of FutureMind. It consistently outperforms strong baselines such as Search-o1, achieving state-of-the-art results under free training conditions across diverse SLM architectures and scales. Beyond empirical gains, our analysis reveals that the process of thinking-pattern distillation is restricted by the cognitive bias bottleneck between the teacher (LLMs) and student (SLMs) models. This provides new perspectives on the transferability of reasoning skills, paving the way for the development of SLMs that combine efficiency with genuine cognitive capability.

## 1 Introduction

In recent years, driven by massive datasets and scalable computing, Large Language Models (LLMs) have achieved outstanding problem-understanding and problem-solving performance on a wide range of general tasks such as commonsense inference (Yang et al., 2025), code generation (Guo et al., 2024), and mathematical reasoning (Shao et al., 2024) through pre-training (Raffel et al., 2020), instruction tuning (Wei et al., 2022a), reinforcement learning from human feedback (RLHF) (Touvron et al., 2023; OpenAI, 2023). However, once problems become time-sensitive or require domain-specific knowledge (Peng et al., 2023; Li et al., 2023b), model performance is constrained by their inherent, static parameters, exposing shortcomings like stale knowledge and insufficient domain coverage. This limitation highlights the necessity of introducing external knowledge sources during reasoning. Against this backdrop, Retrieval-Augmented Generation (RAG)(Gao et al., 2023; Xiong et al., 2025) has emerged: by supplying the model with retrieved documents before inference, it effectively enhances both the accuracy and domain adaptability of language models. Yet, single-step retrieval often struggles with knowledge-intensive, multi-hop reasoning tasks (Yang et al., 2018; Ho et al., 2020b). In response, recent studies have proposed "deep search" paradigms (Li et al., 2025c; Alzubi et al., 2025) that emphasize dynamic interaction between reasoning and retrieval: during problem solving, the model continuously decomposes the question, iteratively retrieves information, and aggregates evidence until the answer converges.

---

*Work done during the internship at XiaoMi.
†Corresponding author.

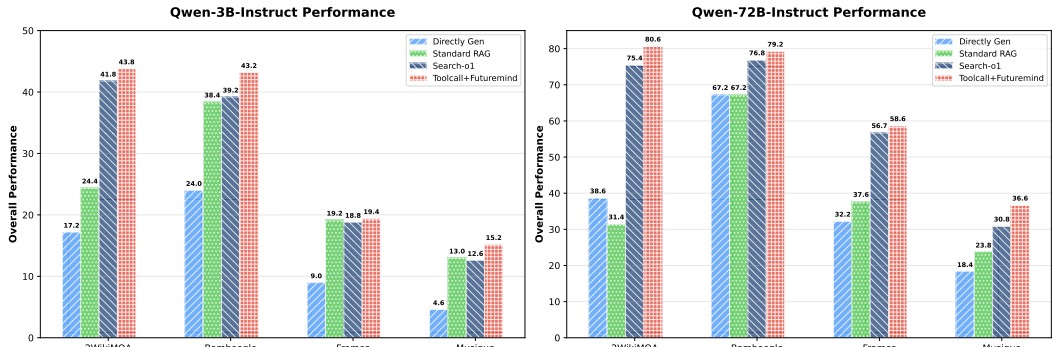

Figure 1: Overall performance comparison of FutureMind with other methods across four multi-hop QA benchmarks. The left panel depicts the performance on a 3B small language model (SLM), while the right panel illustrates the performance on a 72B large language model (LLM).

As problem complexity grows, increasing model size or memory alone is insufficient; effective reasoning also requires explicit "retrieval logic" to determine when, what, and how to retrieve relevant evidence (Schick et al., 2023; Zhang et al., 2024). Search-o1 (Li et al., 2025a) integrates retrieval into the chain-of-thought, while ReAct (Yao et al., 2023) formalizes a "reasoning–acting–observing–reasoning" paradigm for targeted external information (Li et al., 2025b). These approaches illustrate a long-standing consensus: LLM capabilities should be activated dynamically and on demand during inference (Wang et al., 2024b; Jin et al., 2024b). For autonomous agents, this shifts the objective from answering directly to reasoning systematically—analyzing, decomposing, and integrating evidence for deeper insight and more robust strategies.

However, implementing explicit retrieval logic places substantial demands on model capabilities. LLMs are proficient in multi-turn reasoning and retrieval but incur prohibitive latency and computational costs (Wan et al., 2023; Wang, 2024). In contrast, SLMs offer notable advantages in efficiency, cost, and privacy, but their limited memory, weak context retention, and restricted structured reasoning hinder effective problem decomposition, iterative evidence retrieval, and multi-hop aggregation (Wang et al., 2024a; Xu et al., 2025). Consequently, achieving an optimal balance between reasoning effectiveness and computational efficiency remains a critical and unresolved challenge (Bai et al., 2024), particularly for resource-constrained models deployed in real-time or privacy-sensitive scenarios.

To this end, we propose **FutureMind**, a training-free modular reasoning framework that enables low-latency and high-accuracy complex reasoning without gradient updates, leveraging an adaptive thinking-pattern distillation strategy. The name FutureMind reflects our vision for future AI systems: even under constrained resources, the model can draw on distilled thinking-pattern priors to generalize to high-difficulty and unseen problems with free training. FutureMind decomposes reasoning into a four-stage pipeline—**Problem Analysis**, **Logical Reasoning**, **Strategy Planning**, and **Retrieval Guidance**—which sequentially address whether to retrieve, what to retrieve, how to integrate retrieved evidence, and how to generate a coherent answer. To further reduce retrieval overhead, we design three retrieval paradigms based on the decomposition of complex-question retrieval logic: (1) **Forward Stepwise Reasoning** — progressive expansion of sub-queries; (2) **Backward Constraint Focusing** — start from answer constraints and narrow search; (3) **Parallel Intersection Reasoning** — run parallel sub-searches and intersect evidence. By completing the reasoning strategy within a single turn, the framework endows models with clear planning, retrieval, and knowledge-synthesis capabilities, bridging the gap between reasoning depth and efficiency. The contributions of this paper are summarised as follows:

1. **A training-free modular reasoning framework:** We propose FutureMind, a four-stage pipeline (Problem Analysis, Logic Reasoning, Strategy Planning, Retrieval Guidance) supplemented by a dynamic thinking module that provides explicit knowledge support for structured reasoning. The framework is applicable to both LLMs and SLMs, balancing accuracy and efficiency, as shown in Figure 1.

2. **Composable retrieval strategies:** We design three adaptive retrieval paradigms (forward stepwise reasoning, backward constraint focusing, parallel intersection reasoning) that decompose complex multi-hop questions into manageable sub-queries and perform evidence integration efficiently.

3. **Systematic experiments and cognitive insights:** Experiments on four multi-hop QA benchmarks demonstrate that FutureMind consistently improves performance across models of various architectures and scales, with the largest gains on SLMs, establishing a new state of the art among training-free methods. Moreover, we identify a "cognitive-bias bottleneck": once the teacher's plan surpasses the student's capacity, distillation becomes lossy, snapping reasoning chains and amplifying noise. This emphasizes the importance of teacher-student compatibility over raw model size, offering guidance for the design of lightweight yet scalable reasoning systems.

## 2 RELATED WORK

**Large Language Models and Retrieval.** LLMs (Achiam et al., 2023; Team, 2024) exhibit strong reasoning and code-generation abilities (Guo et al., 2025; 2024), but remain prone to hallucination due to their reliance on static parametric knowledge (Zhang et al., 2023). To mitigate this, external search is widely adopted via (i) retrieval-augmented generation (RAG) (Gao et al., 2023), which integrates retrieved evidence into the generation process, and (ii) search-as-a-tool (Schick et al., 2023), where LLMs explicitly interact with a search engine through prompting (Trivedi et al., 2022; Schick et al., 2023) or fine-tuning (Schick et al., 2023). However, RAG's static, single-stage retrieval—i.e., a non-adaptive, one-shot lookup that ignores query complexity and intermediate generation signals—can return irrelevant or weakly informative passages, impeding compositional and multi-hop reasoning (Jin et al., 2024a); tool-based approaches, though more interactive, still struggle to retrieve evidence that is sufficiently relevant and precise for complex, multi-step inference.

**Small Language Models and Cognitive Transfer.** SLMs are attractive for cost-sensitive, low-latency, and privacy-preserving applications, yet they exhibit pronounced deficiencies in memory, context propagation, and structured, multi-step reasoning (Fu et al., 2023; Hsieh et al., 2023). To close this gap, **Cognitive-Transfer techniques** attempt to migrate reasoning behaviors from larger models. **CoT Distillation** transfers step-by-step traces (Wei et al., 2022b; Wang et al., 2023; Fu et al., 2023) but provides limited adaptivity and can be brittle under distributional or stylistic shift. **Prompt Distillation** reduces stylistic mismatch by extracting compact prompts (Li et al., 2023a; Chen & Feng, 2023), yet typically encodes mostly static knowledge templates that do not support dynamic planning. Retrieval-augmented transfers such as **Meta-RAG** improve efficiency through external knowledge (Mombaerts et al., 2024), but commonly treat retrieval as a fixed, non-adaptive pipeline and thus fail to fully integrate retrieval with adaptive reasoning. Overall, these approaches only partially mitigate SLMs' reasoning deficits and lack the generalizability and dynamic adaptivity required for robust problem decomposition, iterative retrieval, and multi-hop aggregation—motivating methods that endow SLMs with lightweight, structured reasoning routines and strategic retrieval policies.

## 3 METHODOLOGY

### 3.1 OVERVIEW

FutureMind is a modular reasoning framework that employs **adaptive knowledge distillation** to transfer structured reasoning and retrieval strategies from teacher models to student models. Unlike conventional distillation methods, which primarily focus on compressing knowledge representations, FutureMind targets the distillation of **systematic thinking patterns**. Specifically, it captures the complete logical chain from problem definition to retrieval guidance, abstracting these patterns into lightweight, reusable strategic thinking-pattern priors. This design enables student models, particularly SLM, to perform adaptive reasoning and deep, structured retrieval planning, thereby achieving superior retrieval performance even in resource-constrained environments.

As depicted in Figure 2, FutureMind is coordinated by the **Thinking Module**, which dynamically generates the optimal retrieval strategies based on task characteristics, data availability, and efficiency

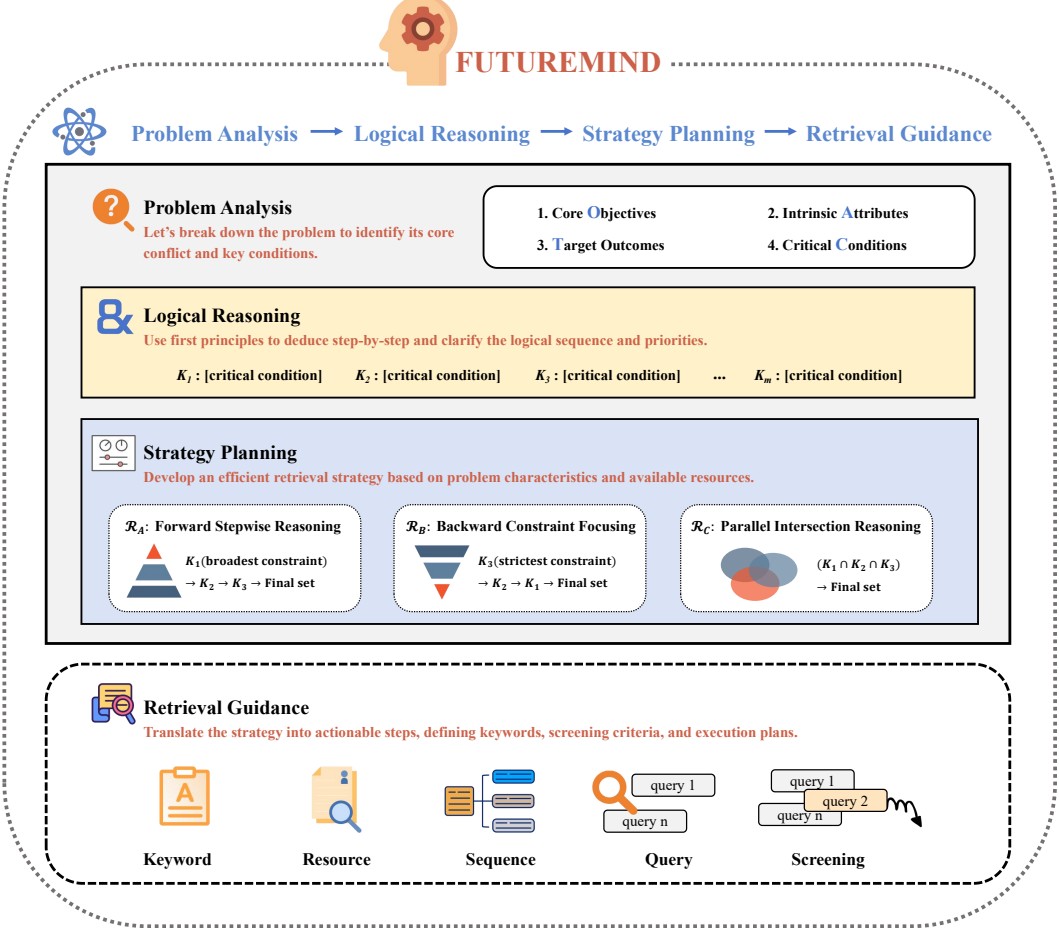

Figure 2: Overview of the FutureMind framework.

constraints. It of four core modules: **Problem Analysis**, **Logical Reasoning**, **Strategy Planning**, and **Retrieval Guidance**, achieving modularity, interpretability, and end-to-end optimization.

Formally, FutureMind is a four-stage pipeline coordinated by the Thinking Module $\mathcal{M}$:

$$F = \mathcal{M}\langle \mathcal{P}, \mathcal{L}, \mathcal{S}, \mathcal{R} \rangle, \tag{1}$$

where $\mathcal{P}$, $\mathcal{L}$, $\mathcal{S}$, and $\mathcal{R}$ represent Problem Analysis, Logical Reasoning, Strategy Planning, and Retrieval Guidance, respectively.

For clarity and reproducibility, we further provide module-wise Instructions and execution examples in the Appendix E.5.1- E.5.4, illustrating how each component operates within the overall framework.

Subsequently, we provide a comprehensive overview of the four core modules of FutureMind.

### 3.2 MODULE DEFINITIONS

#### 3.2.1 PROBLEM ANALYSIS $\mathcal{P}$

The Problem Analysis module initiates the reasoning pipeline by decomposing the input query $x$ into its fundamental components. This decomposition yields a structured representation that enables subsequent reasoning and decision-making processes. Specifically, this module identifies the following key elements:

$$\mathcal{P}(x) \to (\mathcal{O}, \mathcal{A}, \mathcal{T}, \mathcal{C}), \tag{2}$$

where:

- $\mathcal{O}$ represents the core objectives, which define the primary direction and desired outcomes of the problem-solving process.

- $\mathcal{A}$ denotes the intrinsic attributes, characterizing the inherent properties and conditions of the problem.

- $\mathcal{T}$ specifies the target outcomes, indicating the expected results or output types upon problem resolution.

- $\mathcal{C} = \{C_1, C_2, \ldots, C_n\}$ captures the key dimensions, which are critical conditions or factors within the problem, each $C_i$ representing a specific dimension.

By systematically decomposing the input query into these components, the Problem Analysis module establishes a clear, structured foundation for the subsequent reasoning stages.

### 3.2.2 LOGICAL REASONING $\mathcal{L}$

The Logical Reasoning module applies a *first-principles approach* to derive core problem mechanisms. It identifies the most fundamental principles and applies deductive reasoning to construct a coherent abstraction. Unlike heuristic or analogy-based reasoning, this approach grounds inference in causal structures, reducing reliance on incomplete prior knowledge. Formally:

$$\mathcal{L}(\mathcal{O}, \mathcal{A}, \mathcal{T}, \mathcal{C}) \to (\mathcal{M}, \mathcal{K}), \tag{3}$$

where:

- $\mathcal{O}$, $\mathcal{A}$, $\mathcal{T}$, and $\mathcal{C}$ denote the core objectives, intrinsic attributes, target outcomes, and key dimensions extracted from Problem Analysis.

- $\mathcal{M}$ represents the mechanistic understanding, capturing causal relations and fundamental principles governing the system.

- $\mathcal{K} = \{K_1, K_2, \ldots, K_m\}$ is an ordered set of critical conditions, prioritized by logical dependency and discriminative importance.

Grounding reasoning in first principles improves interpretability and adaptability, especially in complex or knowledge-intensive tasks where heuristics are insufficient. By decomposing the problem into basic components, identifying key conditions, and arranging them in order of logical priority, the module converts the structured input from Problem Analysis into a mechanistic abstraction $(\mathcal{M}, \mathcal{K})$. This abstraction serves as the foundation for Strategy Planning and Retrieval Guidance.

### 3.2.3 STRATEGY PLANNING $\mathcal{S}$

The Strategy Planning module bridges the mechanistic insights from Logical Reasoning and the normative layer of Retrieval Guidance. It dynamically determines the optimal retrieval strategy $\mathcal{R}^*$ based on the mechanistic understanding $\mathcal{M}$ and the prioritized condition sequence $\mathcal{K} = \{K_1, K_2, \ldots, K_m\}$. Formally:

$$\mathcal{S}(\mathcal{M}, \mathcal{K}) \to \mathcal{R}^*, \qquad \mathcal{R}^* = \arg\min_{\mathcal{R} \in \mathcal{P}_{\text{cand}}} \mathcal{F}(\mathcal{R}; \mathcal{M}, \mathcal{K}), \tag{4}$$

where:

- $\mathcal{M}$: mechanistic understanding derived from logical reasoning.

- $\mathcal{K} = \{K_1, K_2, \ldots, K_m\}$: prioritized sequence of conditions.

- $\mathcal{P}_{\text{cand}} = \{\mathcal{R}_A, \mathcal{R}_B, \mathcal{R}_C\}$: candidate pool of reasoning strategies.

- $\mathcal{F}(\cdot)$: a cost function evaluating efficiency, constraints, interdependencies, and data availability. In this work, $\mathcal{F}$ specifically refers to the function implemented by the teacher model to conduct a preliminary evaluation of retrieval plans, aiming to generate recommendations for the optimal retrieval strategy.

Based on insights from knowledge-intensive tasks, we design three distinct retrieval paradigms (Figure 4) to enable more efficient and accurate execution. The Strategy Planning module dynamically selects among them based on the condition set topology $\mathcal{K}$.

Let $\mathcal{U}$ denote the candidate space. For each condition $K_i$, we define an evaluation function $\phi(K_i, x) \in \{0, 1\}$ that checks if candidate $x \in \mathcal{U}$ satisfies $K_i$. Intermediate sets are defined as:

$$X_i = \{x \in \mathcal{U} \mid \phi(K_i, x) = 1\}, \qquad X^* = \bigcap_{i=1}^{m} X_i. \tag{5}$$

**Strategy A: Forward Stepwise Reasoning ($\mathcal{R}_A$)**   Applied when early conditions are broad yet effective in pruning. Constraints are applied sequentially from general to specific:

$$X_1 = \{x \in \mathcal{U} \mid \phi(K_1, x) = 1\}, \;\; X_j = \{x \in X_{j-1} \mid \phi(K_j, x) = 1\}, \;\; X^* = \bigcap_{i=1}^{m} X_i. \tag{6}$$

**Strategy B: Backward Constraint Focusing ($\mathcal{R}_B$)**   Adopted when downstream conditions are highly selective. Reasoning starts with the tightest constraint and broadens progressively:

$$X_m = \{x \in \mathcal{U} \mid \phi(K_m, x) = 1\}, \;\; X_j = \{x \in X_{j+1} \mid \phi(K_j, x) = 1\}, \;\; X^* = \bigcap_{i=1}^{m} X_i. \tag{7}$$

**Strategy C: Parallel Intersection Reasoning ($\mathcal{R}_C$)**   Best suited for independent or orthogonal conditions. All constraints are processed in parallel, then intersected:

$$X_i = \{x \in \mathcal{U} \mid \phi(K_i, x) = 1\}, \qquad X^* = \bigcap_{i=1}^{m} X_i. \tag{8}$$

By deeply understanding knowledge-intensive problems, the module adaptively selects the optimal retrieval strategy, ensuring both efficient and precise retrieval execution.

### 3.2.4  RETRIEVAL GUIDANCE $\mathcal{R}$

The Retrieval Guidance module serves as a normative layer that transforms abstract reasoning and the selected retrieval strategy into structured instructions for execution. Unlike direct retrieval, this module generates prescriptive guidelines that specify how retrieval should be performed.

Given the mechanistic understanding $\mathcal{M}$, the prioritized conditions $\mathcal{K}$, and the chosen strategy $\mathcal{R}^*$, the module outputs a set of retrieval guidelines:

$$\mathcal{R}(\mathcal{M}, \mathcal{K}, \mathcal{R}^*) \to \Gamma, \tag{9}$$

where $\Gamma = \{\gamma_1, \gamma_2, \dots, \gamma_q\}$ is a set of normative principles guiding the retrieval process(e.g., priority order, source preferences, evaluation criteria).

The guidance is structured into five key, complementary stages:

- **Keyword Guidance.** Extract core entities, attributes, and relations from $\mathcal{K}$ and specify the lexical and semantic variants that retrieval should prioritize. This guidance outlines the dimensions along which queries can vary, enabling adaptive retrieval across domains while maintaining alignment with the underlying reasoning structure.

- **Resource Guidance.** Indicate categories of information sources (e.g., academic databases, industry reports, policy documents) ranked by relevance $\mathcal{M}$ and credibility, guiding retrieval toward reliable knowledge domains.

- **Sequence Guidance.** Provide recommendations on the ordering of retrieval steps in accordance with $\mathcal{R}^*$. For instance, a Forward Stepwise strategy begins with broad, high-recall repositories to establish initial coverage before moving to domain-specific collections. Conversely, a Backward Constraint strategy starts with highly selective regulatory to anchor the search with high-precision evidence, then expands outward as needed.

- **Query Guidance.** Provide structural templates for query formulation (e.g., Boolean patterns, semantic expansions, hierarchical constraints), emphasizing inclusiveness in initial searches and progressive narrowing in later stages. This guidance offers adaptable design principles rather than fixed query strings.

- **Screening Guidance.** Define the principles for evaluating retrieved results, including their relevance to $\mathcal{M}$, source credibility, and methodological rigor. The module specifies evaluation criteria conceptually.

By structuring guidance around keywords, resources, sequencing, query formulation, and evaluation, the module bridges cognitive strategy with retrieval while maintaining executional independence.

## 4 EXPERIMENTS

### 4.1 EXPERIMENTAL SETUP

**Datasets.** We evaluate our approach on four widely-used multi-hop question answering (QA) benchmarks: **2WikiMultihopQA (2WikiMQA)** (Ho et al., 2020a), **Bamboogle** (Press et al., 2022), **MuSiQue** (Tang & Yang, 2024), and **FRAMES** (Krishna et al., 2024). Specifically, we randomly sample 500 instances from the validation sets of 2WikiMQA and MuSiQue, while evaluating on the full test sets of Bamboogle and FRAMES.

**Metrics.** For evaluation, following prior work (Sun et al., 2025), we adopt two complementary metrics: **Coverage-based Exact Match (ACC$_E$)** and **LLM-as-Judge (ACC$_L$)**. ACC$_E$ measures whether the predicted answer fully covers the gold reference while allowing for paraphrastic variations; its detailed calculation formula is provided in Appendix G.1. In contrast, ACC$_L$ employs GPT-4o-mini as an automatic evaluator to judge the semantic correctness of predicted answers relative to the gold reference. The full evaluation prompt for ACC$_L$ is provided in Appendix F.1.

**Baselines.** We consider three categories of baselines: (1) **Naive Generation**: Generates answers without retrieval. (2) **Standard RAG** (Zhao et al., 2024): Retrieves documents using the original question as the query. (3) **Search-o1** (Li et al., 2025a): Performs self-initiated retrieval using prompts.

**Implementation Details.** We evaluate the effectiveness of the proposed **FutureMind** method using models at different architectures and scales (Qwen-2.5-3B/7B/14B/32B/72B-Instruct and Llama3.1-8B-Instruct). For generation, we set the maximum sequence length to 32768 tokens, with temperature = 0.0, top-p = 0.8, top-k = 20, and repetition penalty = 1.05 across all models. For retrieval, we employ the Google Web Search API, retrieving the top k = 10 results. In experiments, FutureMind leverages an enhanced version of **Toolcall (TC)**, a ReAct-Style orchestration framework [*]. We modify the original TC framework by replacing its single search process with parallel search, enabling more efficient and robust aggregation of retrieved evidence. This configuration is referred to as **TC+FM**. Details of implementation are provided in Appendix E.

### 4.2 MAIN RESULTS

Table 1 compares the performance of different model architectures and scales across four multi-hop QA benchmarks, under four methods: naive generation, standard RAG, Search-o1, and ToolCall-driven FutureMind. Several key observations emerge from the results. Additional benchmark results are presented in Table 6 in Appendix B due to space limitations.

1. **Inherent Limitations of Baseline Methods.** Naive generation (internal knowledge only) yields the lowest accuracy, underscoring its inability to integrate external evidence. While standard RAG (retrieval-enabled) cannot reliably perform multi-step reasoning and may even underperform naive generation when reasoning integration fails. Search-o1 (reason-in-documents) enhances retrieval quality but remains limited: small models benefit minimally, and even larger models are constrained by these intrinsic summarization and integration capacity.

2. **Effectiveness of FutureMind with Adaptive Knowledge Distillation.** Unlike Search-o1's fixed-prompt design, FutureMind employs adaptive knowledge, enabling a more flexible and effective problem-solving process that yields consistent performance gains. For instance, Qwen-3B improves on Frames (ACC$_E$: 11.77 → 18.84), Llama3.1-8B on Bamboogle (ACC$_L$: 52.00 → 64.00), and Qwen-72B on 2WikiMQA (ACC$_L$: 75.40 → 80.60). By providing adaptive external

---

[*]https://github.com/QwenLM/Qwen-Agent/

Table 1: Main results on four multi-hop QA benchmarks. **Bold** denotes the best performance and underline indicates the second best. Rows with blue background correspond to our methods TC+FM*, where FM* selects the best-performing FutureMind-enhanced variant for each base model.

| Model | Method | 2WikiMQA | | Bamboogle | | Frames | | MuSiQue | | AVG | |
|---|---|---|---|---|---|---|---|---|---|---|---|
| | | $ACC_E$ | $ACC_L$ | $ACC_E$ | $ACC_L$ | $ACC_E$ | $ACC_L$ | $ACC_E$ | $ACC_L$ | $ACC_E$ | $ACC_L$ |
| Qwen-3B | Naive Gen | 16.80 | 17.20 | 20.80 | 24.00 | 5.94 | 8.98 | 3.60 | 4.60 | 11.79 | 13.70 |
| | Standard RAG | 24.00 | 24.40 | 26.40 | 38.40 | 12.01 | 19.17 | 10.20 | 13.00 | 18.15 | 23.74 |
| | Search-o1 | 41.00 | 41.80 | 34.40 | 39.20 | 11.77 | 18.81 | 10.40 | 12.60 | 24.39 | 28.10 |
| | TC+FM* | **56.40** | **43.80** | **39.20** | **43.20** | **18.84** | **19.42** | **14.20** | **15.20** | **32.16** | **30.41** |
| Qwen-7B | Naive Gen | 29.40 | 25.20 | 34.40 | 37.60 | 11.29 | 16.87 | 7.60 | 10.80 | 20.67 | 22.62 |
| | Standard RAG | 30.20 | 29.80 | 42.40 | 52.80 | 15.78 | 24.76 | 13.20 | 16.80 | 25.39 | 31.04 |
| | Search-o1 | 57.80 | 59.80 | 43.20 | 51.20 | 24.63 | **38.34** | **20.80** | **23.80** | 36.61 | 43.29 |
| | TC+FM* | **62.00** | **64.00** | **58.40** | **64.80** | **25.12** | 34.71 | 20.00 | **23.80** | **41.38** | **46.83** |
| Qwen-14B | Naive Gen | 30.40 | 30.80 | 48.80 | 55.20 | 14.81 | 22.82 | 8.80 | 12.40 | 25.70 | 30.30 |
| | Standard RAG | 27.40 | 28.40 | 44.80 | 56.00 | 17.96 | 28.40 | 14.00 | 18.60 | 26.04 | 32.85 |
| | Search-o1 | 66.80 | 68.40 | 43.20 | 55.20 | 30.46 | 46.48 | 20.60 | 25.60 | 40.27 | 48.92 |
| | TC+FM* | **71.60** | **75.20** | **70.40** | **72.80** | **34.83** | **49.51** | **24.00** | **28.20** | **50.21** | **56.43** |
| Qwen-32B | Naive Gen | 30.80 | 31.30 | 54.40 | 60.80 | 15.66 | 24.51 | 10.80 | 15.20 | 27.91 | 32.95 |
| | Standard RAG | 24.60 | 24.40 | 52.80 | 61.60 | 19.78 | 30.95 | 16.20 | 19.60 | 28.35 | 34.14 |
| | Search-o1 | 68.60 | 71.60 | 60.80 | 67.20 | 34.34 | **54.12** | 22.80 | 27.80 | 46.63 | 55.18 |
| | TC+FM* | **74.40** | **77.80** | **68.80** | **72.80** | **37.15** | 53.86 | **26.00** | **30.40** | **51.59** | **58.71** |
| Qwen-72B | Naive Gen | 38.20 | 38.60 | 60.00 | 67.20 | 21.12 | 32.16 | 12.80 | 18.40 | 33.03 | 39.09 |
| | Standard RAG | 31.00 | 31.40 | 59.20 | 67.20 | 25.97 | 37.62 | 19.00 | 23.80 | 33.79 | 40.01 |
| | Search-o1 | 72.60 | 75.40 | 67.20 | 72.80 | 37.37 | 56.67 | 24.60 | 30.80 | 50.44 | 58.92 |
| | TC+FM* | **74.20** | **80.60** | **75.20** | **79.20** | **41.38** | **58.59** | **28.40** | **36.60** | **54.80** | **63.75** |
| Llama3.1-8B | Naive Gen | 38.20 | 38.60 | **60.00** | **67.20** | 21.12 | 32.16 | 12.80 | 18.40 | 33.03 | 39.09 |
| | Standard RAG | 29.20 | 30.40 | 39.20 | 47.20 | 15.05 | 22.82 | 12.20 | 15.20 | 23.91 | 28.90 |
| | Search-o1 | 54.00 | 56.00 | 46.40 | 52.00 | 24.88 | 37.62 | 15.40 | 18.20 | 35.17 | 40.95 |
| | TC+FM* | **55.20** | 56.80 | 58.40 | 64.00 | **27.43** | **39.92** | **21.80** | **25.20** | **40.71** | **46.48** |

Table 2: Impact of Teacher Model Scale on Student Performance in multi-hop QA benchmarks. **Bold** denotes the best performance and underline indicates the second best. Rows with blue background correspond to the best teacher model scale for each student model.

| Model | Method | 2WikiMQA | | Bamboogle | | Frames | | MuSiQue | | Avg | |
|---|---|---|---|---|---|---|---|---|---|---|---|
| | | $ACC_E$ | $ACC_L$ | $ACC_E$ | $ACC_L$ | $ACC_E$ | $ACC_L$ | $ACC_E$ | $ACC_L$ | $ACC_E$ | $ACC_L$ |
| Qwen-3B | TC | 54.20 | 42.40 | 37.60 | 40.00 | 17.96 | **22.94** | 13.00 | 12.60 | 30.69 | 29.49 |
| | TC+FM (3B) | 42.00 | 30.80 | 28.00 | 30.40 | 11.53 | 10.32 | 7.40 | 8.60 | 22.23 | 20.03 |
| | TC+FM (7B) | 53.00 | 39.60 | 30.40 | 36.00 | 15.78 | 16.26 | 11.20 | 11.40 | 27.60 | 25.82 |
| | TC+FM (14B) | 55.20 | **45.60** | **40.80** | 42.40 | 17.11 | 18.46 | 12.20 | **12.80** | **31.33** | 29.82 |
| | TC+FM (32B) | 49.20 | 37.00 | 36.00 | 37.60 | 12.26 | 12.01 | 10.20 | 10.40 | 26.92 | 24.25 |
| | TC+FM (72B) | **56.40** | 43.80 | 39.20 | **43.20** | 17.84 | 19.42 | **14.20** | 15.20 | 31.91 | 30.41 |
| Qwen-7B | TC | 56.80 | 56.20 | 49.60 | 54.40 | 23.78 | 32.28 | 16.40 | 18.80 | 36.65 | 40.42 |
| | TC+FM (3B) | 60.20 | 60.00 | 49.60 | 50.40 | 23.09 | 30.83 | 17.00 | 19.60 | 37.97 | 40.21 |
| | TC+FM (7B) | 60.20 | 61.80 | 53.60 | 57.60 | 24.39 | 33.55 | 17.60 | 21.20 | 38.95 | 43.04 |
| | TC+FM (14B) | **62.00** | **64.00** | **58.40** | **64.80** | 25.12 | **34.71** | **20.00** | **23.80** | **41.38** | **46.83** |
| | TC+FM (32B) | 57.80 | 57.60 | 52.00 | 58.40 | 22.57 | 29.98 | 15.20 | 19.40 | 36.89 | 41.35 |
| | TC+FM (72B) | 60.40 | 60.00 | 56.80 | 61.60 | **26.58** | **34.71** | 18.20 | 21.20 | 40.50 | 44.38 |

strategy empowerment rather than depending solely on internal capability, FutureMind achieves stronger and more scalable reasoning, particularly under resource-constrained settings.

3. **Universal Applicability of FutureMind Across Model Architectures and Scales.** TC+FM* achieves state-of-the-art results in nearly all settings, delivering scalable improvements across both model architectures and parameter scales. This demonstrates FutureMind's broad effectiveness in enhancing multi-hop reasoning via external strategy transfer, alleviating capability bottlenecks in resource-limited models, while maintaining strong performance in larger models.

Table 3: Ablation study of modular components on multi-hop QA benchmarks. **Bold** denotes the performance with all modules enabled.

| Model | Method | 2WikiMQA | | Bamboogle | | Frames | | MuSiQue | | Avg | |
|-------|--------|----------|----------|-----------|----------|--------|--------|---------|---------|-------|-------|
| | | $ACC_E$ | $ACC_L$ | $ACC_E$ | $ACC_L$ | $ACC_E$ | $ACC_L$ | $ACC_E$ | $ACC_L$ | $ACC_E$ | $ACC_L$ |
| | All Modules | **62.00** | **64.00** | **58.40** | **64.80** | **25.12** | **34.71** | **20.00** | **23.80** | **41.38** | **46.83** |
| | - w/o Problem Analysis | 58.20 | 57.40 | 56.00 | 62.40 | 24.93 | 34.57 | 16.00 | 21.00 | 38.78 | 43.84 |
| Qwen-7B | - w/o Logical Reasoning | 60.40 | 59.00 | 49.60 | 53.60 | 24.59 | 34.21 | 19.20 | 23.00 | 38.45 | 42.45 |
| | - w/o Strategy Planning | 56.40 | 57.20 | 49.30 | 51.20 | 22.63 | 33.10 | 17.60 | 19.80 | 36.48 | 40.33 |
| | - w/o Retrieval Guidance | 59.20 | 60.40 | 57.60 | 60.00 | 23.39 | 33.34 | 18.40 | 20.60 | 39.65 | 43.59 |

Table 4: Ablation study of retrieval strategies on multi-hop QA benchmarks. **Bold** denotes the performance with all three retrieval strategies distilled from the Qwen-14B teacher.

| Model | Method | 2WikiMQA | | Bamboogle | | Frames | | MuSiQue | | Avg | |
|-------|--------|----------|----------|-----------|----------|--------|--------|---------|---------|-------|-------|
| | | $ACC_E$ | $ACC_L$ | $ACC_E$ | $ACC_L$ | $ACC_E$ | $ACC_L$ | $ACC_E$ | $ACC_L$ | $ACC_E$ | $ACC_L$ |
| | All Strategies | **62.00** | **64.00** | **58.40** | **64.80** | **25.12** | **34.71** | **20.00** | **23.80** | **41.38** | **46.83** |
| Qwen-7B | - w/o Strategy A | 57.40 | 57.80 | 54.20 | 60.00 | 24.64 | 32.16 | 16.60 | 21.20 | 38.21 | 42.79 |
| | - w/o Strategy B | 58.80 | 58.00 | 57.60 | 61.60 | 25.09 | 34.47 | 18.40 | 22.60 | 39.97 | 44.67 |
| | - w/o Strategy C | 60.40 | 59.00 | 54.40 | 60.80 | 24.72 | 33.37 | 17.40 | 21.40 | 39.23 | 43.64 |

## 4.3 IMPACT OF TEACHER MODEL DESIGN ON TEACHER–STUDENT COGNITIVE ALIGNMENT

We systematically analyze the impact of teacher model design on student performance in knowledge distillation (Table 2). Several consistent patterns emerge:

1. **Small-scale teachers models degrade student performance.** In TC+FM, using a 3B teacher for Qwen-3B reduces average performance ($ACC_E$: 30.69 → 22.23, $ACC_L$: 29.49 → 20.03), indicating that low-capacity teachers may generate noisy or misleading planning signals, hindering transfer effectiveness.

2. **Mid-scale teachers provide optimal alignment.** Both student models benefit most from the 14B teacher(Qwen-3B: $ACC_E$: 31.33, $ACC_L$: 29.82; Qwen-7B: $ACC_E$: 41.38, $ACC_L$: 46.83), outperforming the 32B variant and matching or exceeding the 72B model on average.

3. **Cognitive compatibility outweighs raw scale.** Although the 72B teacher excels in certain sub-tasks, it does not consistently surpass the 14B teacher on average (Qwen-7B student: 72B teacher $ACC_E$: 40.50, $ACC_L$:44.38 < 14B teacher $ACC_E$: 41.38, $ACC_L$: 46.83), suggesting that cognitive alignment between teacher and student plays a more critical role in distillation effectiveness than raw scale.

The "cognitive bias bottleneck" further demonstrates that overly complex teacher plans may fail to transfer reasoning capabilities to smaller students, as strategic information loss can disrupt critical reasoning chains or amplify noise. Therefore, in knowledge distillation, prioritizing teacher–student compatibility is more important than considering raw model size. Future work should systematically quantify planning quality and evaluate generalization across tasks and alignment strategies to ensure scalable reasoning in lightweight models.

Beyond scale, the teacher's architecture, reasoning orientation, and instruction tuning must be compatible with the student. Experiments in Appendix B show that teachers with architectures well-aligned to the student consistently enable more effective thinking-pattern distillation, leading to improved multi-hop reasoning performance.

## 4.4 ABLATION STUDIES

We first evaluate the contributions of the four core modules of FutureMind: Problem Analysis, Logical Reasoning, Strategy Planning, and Retrieval Guidance. As shown in Table 3, removing any module leads to noticeable performance drops, confirming their complementary roles. Among them, Strategy Planning has the largest impact, highlighting its central role in converting structured reasoning into effective retrieval actions.

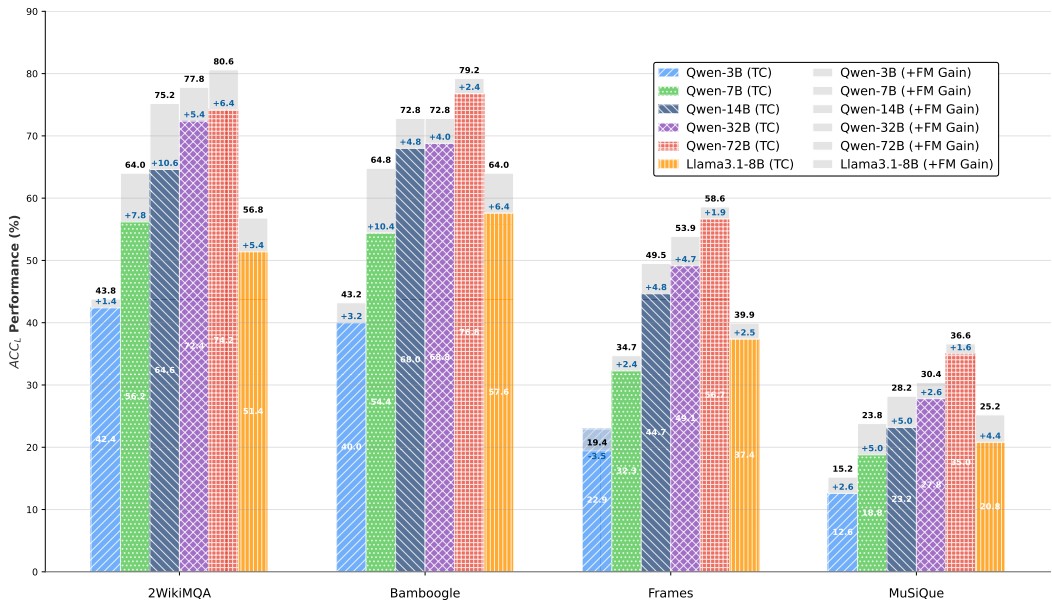

Figure 3: Ablation study of enhanced ToolCall across different models.

Next, we analyze the three retrieval strategies within the Strategy Planning module. Table 4 shows that removing any single strategy—Forward Stepwise Reasoning ($\mathcal{R}_A$), Backward Constraint Focusing ($\mathcal{R}_B$), or Parallel Intersection Reasoning ($\mathcal{R}_C$)—degrades performance. Overall, $\mathcal{R}_A$ is most critical, $\mathcal{R}_C$ contributes more on datasets with independent conditions, and $\mathcal{R}_B$ benefits constraint-focused cases.

Finally, we evaluate the overall impact of FutureMind by comparing enhanced ToolCall alone versus ToolCall-driven FutureMind. Figure 3 shows that removing the FutureMind module consistently lowers performance, demonstrating its essential role in coordinating multi-hop reasoning and retrieval planning. The detailed numerical results are provided in Table 7.

Together, these ablations show that FutureMind, along with its modules and retrieval strategies, establishes an effective and coherent reasoning framework.

## 5 CONCLUSION

We introduce **FutureMind**, a training-free modular reasoning framework that enables both large and small language models to perform efficient, accurate, and structured reasoning. By decomposing reasoning into **Problem Analysis**, **Logical Reasoning**, **Strategy Planning**, and **Retrieval Guidance**, and leveraging adaptive retrieval strategies, FutureMind provides clear guidance on when, what, and how to retrieve evidence, effectively balancing reasoning depth and efficiency.

Evaluations on four multi-hop QA benchmarks show consistent gains across model scales and architectures, with the largest improvements in SLMs. We further identify a *cognitive bias bottleneck* in teacher-student thinking-pattern distillation: overly complex teacher plans can overwhelm student capacity, causing strategic information loss and degraded reasoning. Effective distillation requires both scale and architectural alignment, as structurally compatible teachers with aligned reasoning orientations enable more effective strategy transfer.

In summary, FutureMind shows that structured, adaptive reasoning is achievable even for SLMs, turning them into cognitively capable agents through strategic thinking-pattern priors.

ETHICS STATEMENT

We strictly adhere to the ICLR Code of Ethics. This work only uses four publicly available multi-hop QA datasets and does not involve any personal or sensitive information, nor does it recruit human subjects. There are no conflicts of interest or foreseeable ethical risks associated with this study. Our research does not introduce ethical concerns beyond the scope of standard multi-hop question answering tasks.

REPRODUCIBILITY STATEMENT

We will publicly release all experimental code and data processing scripts upon paper acceptance to ensure reproducibility. The four datasets used are publicly available (e.g., 2WikiMultihopQA, MuSiQue, Bamboogle, Frames), and the appendix details data preprocessing, retrieval configurations, and relevant hyperparameters. The main text provides sufficient descriptions of the model architecture, reasoning pipeline, and evaluation metrics to allow other researchers to replicate our results.

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

## A    THE USE OF LARGE LANGUAGE MODELS (LLMS)

During the writing process of this paper, we utilized the large language models (LLMs) as an auxiliary tool, solely for polishing the grammar and expression of the paper to enhance its standardization and readability. The research conception, core content, experimental design, and conclusions of the paper were all independently completed by the authors, with the large language models not participating in any substantive aspects of the research or creative process.

## B    ADDITIONAL EXPERIMENTAL RESULTS

As shown in Table 5, teacher architecture has a decisive impact on student adaptation under the TC+FM framework: across all scales (3B–72B), Qwen2.5-72B consistently surpasses Llama3.1-70B in both $ACC_E$ and $ACC_L$. This pattern demonstrates that architectural alignment is the key of effective cognitive transfer in multi-hop reasoning.

## C    TOOLS DESIGN

To enable efficient and accurate retrieval-driven reasoning, we adopt the ReAct framework as implemented in Qwen-Agent and develop two complementary tools. The first is a **Parallel Search Tool** C.1 built on the Google Search API that performs parallel retrievals and makes retrieval decisions, enabling high-efficiency searching. The second is **FutureMind Tool** C.2, a module instantiated with a larger language model that strengthens structured reasoning and retrieval logic, enabling more precise, targeted retrieval. Together, these components decouple the complex retrieval process from the reasoning logic, enabling efficient and accurate retrieval execution, thereby facilitating better evidence integration and yielding more accurate final answers.

Table 5: Effect of Teacher Model Architecture on Student Performance. We evaluate student models (3B–72B) under different teacher model architectures, specifically Qwen2.5-72B-Instruct (denoted as Q2.5) and Llama3.1-70B-Instruct (denoted as L3.1) within the Toolcall-driven FutureMind (TC+FM). **Bold** denotes the best performance and underline indicates the second best. Rows with blue background correspond to the best teacher model architecture for each student model.

| Model | Method | 2WikiMQA | | Bamboogle | | Frames | | MuSiQue | | Avg | |
|---|---|---|---|---|---|---|---|---|---|---|---|
| | | $ACC_E$ | $ACC_L$ | $ACC_E$ | $ACC_L$ | $ACC_E$ | $ACC_L$ | $ACC_E$ | $ACC_L$ | $ACC_E$ | $ACC_L$ |
| Qwen-3B | TC | 54.20 | 42.40 | 37.60 | 40.00 | 17.96 | 22.94 | 13.00 | 12.60 | 30.69 | 29.49 |
| | TC+FM (L3.1) | 47.80 | 34.00 | 27.20 | 27.20 | 11.53 | 9.83 | 10.20 | 10.40 | 24.18 | 20.36 |
| | TC+FM (Q2.5) | 56.40 | 43.80 | 39.20 | 43.20 | 17.84 | 19.42 | 14.20 | 15.20 | **31.91** | **30.41** |
| Qwen-7B | TC | 56.80 | 56.20 | 49.60 | 54.40 | 23.78 | 32.28 | 16.40 | 18.80 | 36.65 | 40.42 |
| | TC+FM (L3.1) | 53.60 | 50.20 | 48.80 | 51.20 | 19.54 | 26.58 | 14.20 | 17.20 | 34.03 | 36.30 |
| | TC+FM (Q2.5) | 57.80 | 57.60 | 52.00 | 58.40 | 22.57 | 29.98 | 15.20 | 19.40 | **36.89** | **41.34** |
| Qwen-14B | TC | 64.40 | 64.60 | 64.00 | 68.00 | 34.22 | 44.66 | 20.40 | 23.20 | 45.76 | 50.11 |
| | TC+FM (L3.1) | 63.40 | 63.80 | 63.60 | 67.20 | 28.16 | 37.86 | 22.80 | 25.20 | 44.49 | 48.52 |
| | TC+FM (Q2.5) | 70.00 | 71.60 | 64.00 | 68.00 | 35.92 | 48.06 | 25.80 | 28.20 | **48.93** | **53.96** |
| Qwen-32B | TC | 71.20 | 72.40 | 66.40 | 68.80 | 36.28 | 49.15 | 24.60 | 27.80 | 49.62 | 54.54 |
| | TC+FM (L3.1) | 63.40 | 61.40 | 66.40 | 68.00 | 28.79 | 38.74 | 23.20 | 26.20 | 45.45 | 48.59 |
| | TC+FM (Q2.5) | 74.40 | 77.80 | 68.80 | 72.80 | 34.15 | 47.86 | 26.00 | 30.40 | **50.84** | **57.21** |
| Qwen-72B | TC | 71.60 | 74.20 | 68.80 | 76.80 | 40.04 | 56.67 | 27.40 | 35.00 | 51.96 | 60.67 |
| | TC+FM (L3.1) | 67.40 | 69.80 | 69.60 | 76.00 | 33.98 | 47.33 | 23.20 | 29.60 | 48.54 | 55.68 |
| | TC+FM (Q2.5) | 74.20 | 80.60 | 75.20 | 79.20 | 41.38 | 58.59 | 27.40 | 34.60 | **54.55** | **63.25** |

Table 6: Additional results on four multi-hop QA benchmarks.

| Model | Method | 2WikiMQA | | Bamboogle | | Frames | | MuSiQue | | AVG | |
|---|---|---|---|---|---|---|---|---|---|---|---|
| | | $ACC_E$ | $ACC_L$ | $ACC_E$ | $ACC_L$ | $ACC_E$ | $ACC_L$ | $ACC_E$ | $ACC_L$ | $ACC_E$ | $ACC_L$ |
| Qwen-14B | TC | 64.40 | 64.60 | 64.00 | 68.00 | 34.22 | 44.66 | 20.40 | 23.20 | 45.76 | 50.12 |
| | TC+FM (3B) | 68.40 | 70.00 | 58.40 | 64.00 | 32.89 | 43.33 | 22.80 | 27.00 | 45.62 | 51.08 |
| | TC+FM (7B) | 69.00 | 71.20 | 63.20 | 72.00 | 33.37 | 46.84 | 20.60 | 24.40 | 46.54 | 53.61 |
| | TC+FM (14B) | 71.60 | 75.20 | 70.40 | 72.80 | 34.83 | 49.51 | 24.00 | 28.20 | 50.21 | 56.43 |
| | TC+FM (32B) | 71.80 | 74.40 | 64.80 | 72.00 | 35.25 | 46.24 | 22.20 | 27.20 | 48.51 | 54.96 |
| | TC+FM (72B) | 70.00 | 71.60 | 64.00 | 68.00 | 35.92 | 48.06 | 25.80 | 28.20 | 48.93 | 53.96 |
| Qwen-32B | TC | 71.20 | 72.40 | 66.40 | 68.80 | 36.28 | 49.15 | 24.60 | 27.80 | 49.62 | 54.54 |
| | TC+FM (3B) | 70.40 | 70.00 | 60.80 | 67.20 | 32.77 | 44.53 | 22.60 | 24.80 | 46.64 | 51.63 |
| | TC+FM (7B) | 71.80 | 73.20 | 61.60 | 67.80 | 34.83 | 45.14 | 22.90 | 25.80 | 47.28 | 53.49 |
| | TC+FM (14B) | 74.60 | 76.40 | 70.40 | 75.20 | 35.19 | 47.33 | 23.20 | 29.80 | 50.85 | 57.18 |
| | TC+FM (32B) | 69.60 | 70.00 | 64.00 | 73.60 | 31.10 | 42.89 | 21.00 | 24.60 | 46.43 | 52.77 |
| | TC+FM (72B) | 74.40 | 77.80 | 68.80 | 72.80 | 37.15 | 53.86 | 26.00 | 30.40 | 51.59 | 59.71 |
| Qwen-72B | TC | 71.60 | 74.20 | 68.80 | 76.80 | 40.04 | 56.67 | 27.40 | 35.00 | 51.96 | 60.67 |
| | TC+FM (3B) | 71.20 | 75.20 | 65.60 | 72.00 | 38.23 | 53.64 | 24.60 | 29.40 | 49.91 | 57.56 |
| | TC+FM (7B) | 72.80 | 76.80 | 71.20 | 76.00 | 37.13 | 55.70 | 24.60 | 30.40 | 51.93 | 59.73 |
| | TC+FM (14B) | 71.00 | 73.40 | 74.40 | 77.60 | 35.56 | 50.12 | 23.60 | 30.00 | 51.14 | 57.78 |
| | TC+FM (72B) | 74.20 | 80.60 | 75.20 | 79.20 | 41.38 | 58.59 | 28.40 | 36.60 | 54.80 | 63.75 |

## C.1 PARALLEL SEARCH TOOL

> **Parallel Search Tool (Google) Introduction**
>
> **Name:** Parallel_Search (google)
>
> **Description:**
> You should invoke the Parallel_Search Tool (google) whenever the user's query falls into one of the following categories:
> 1. Your internal knowledge base and training data are insufficient to answer the question accurately.
> 2. The user asks about a specific example, product, or piece of information that you can

Table 7: Ablation study of enhanced ToolCall across different models.

| Method | Model | 2WikiMQA | | Bamboogle | | Frames | | MuSiQue | | Avg | |
|---|---|---|---|---|---|---|---|---|---|---|---|
| | | $ACC_E$ | $ACC_L$ | $ACC_E$ | $ACC_L$ | $ACC_E$ | $ACC_L$ | $ACC_E$ | $ACC_L$ | $ACC_E$ | $ACC_L$ |
| ToolCall | Qwen-3B | 54.20 | 42.40 | 37.60 | 40.00 | 17.96 | 22.94 | 13.00 | 12.60 | 30.69 | 29.49 |
| | Qwen-7B | 56.80 | 56.20 | 49.60 | 54.40 | 23.78 | 32.28 | 16.40 | 18.80 | 36.65 | 40.42 |
| | Qwen-14B | 64.40 | 64.60 | 64.00 | 68.00 | 34.22 | 44.66 | 20.40 | 23.20 | 45.76 | 50.11 |
| | Qwen-32B | 71.20 | 72.40 | 66.40 | 68.80 | 36.28 | 49.15 | 24.60 | 27.80 | 49.62 | 54.54 |
| | Qwen-72B | 71.60 | 74.20 | 68.80 | 76.80 | 40.04 | 56.67 | 27.40 | 35.00 | 51.96 | 60.67 |
| | Llama3.1-8B | 51.20 | 51.40 | 55.20 | 57.60 | 24.51 | 37.38 | 17.40 | 20.80 | 37.08 | 41.80 |

retrieve in greater detail via the web.

3. The question involves the latest data, dynamic information, or any knowledge that postdates your training cutoff and requires real-time updates.

4. The answer exists in external knowledge sources you cannot directly access; you must search to retrieve it.

5. Although you possess general knowledge of the topic, an online search would yield more detailed or up-to-date information (e.g. current buzzwords or trending topics).

6. You encounter an unfamiliar term or concept and must avoid fabrication by verifying it through the search tool.

7. You need to consult a product manual or official specification to support your response.

The search tool supports both parallel and sequential queries:

1. If multiple searches are independent, you may issue them in parallel.

2. If queries depend on each other (i.e. require ordered steps), perform them sequentially.

**Parameters:**

```
{
  "type": "object",
  "properties": {
    "queries": {
      "type": "array",
      "items": { "type": "string" },
      "description": (
        "List of search keywords:"
        "- Parallel search: supply multiple keywords at once;"
        "- Iterative search: supply a single-element array."
      ),
      "examples": [
        {
          "queries": [
            "Xiaomi SU7 Ultra official price",
            "Tesla Model S latest price"
          ]
        },
        {
          "queries": ["Mishi wolffin fish namer"]
        }
      ]
    }
  },
  "required": ["queries"]
}
```

## C.2 FUTUREMIND TOOL

---

**FutureMind Tool Instroduction**

**Name:** FutureMind

**Description:**
Upon receiving a query, the FutureMind Tool is invoked first to obtain **a systematic thinking pattern and solution roadmap** for the problem.

For retrieval-oriented questions:
1. It produces a structured problem-solving workflow and retrieval strategy.
2. It explicitly delineates the logical chain from "Problem Definition" through "Condition Decomposition" to "Conclusion Derivation."
3. It provides executable search sequences and combined query conditions as retrieval guidance, thereby enhancing information acquisition efficiency and ensuring retrieval accuracy.

**Parameters:**

```
{
    "type": "object",
    "properties": {
        "query": {
            "type": "string",
            "description": "A query that requires systematic
                problem analysis and retrieval-strategy formulation"
            }
    },
    "required": ["query"]
}
```

---

## D TOOLCALL TEMPLATE

---

**Toolcall Template**

**Tools**
You may call one or more functions to assist with the user query.

You are provided with function signatures within <tools></tools> XML tags:

```
<tools>
{tool_descs}
</tools>
```

For each function call, return a json object with function name and arguments within <tool_call></tool_call> XML tags:

```
<tool_call>
{"name": <function-name>, "arguments": <args-json-object>}
</tool_call>
```

---

## E FUTUREMIND: DESCRIPTION AND IMPLEMENTATION DETAILS

### E.1 OVERVIEW OF FUTUREMIND

As depicted in Figure 2, FutureMind is a lightweight, training-free reasoning framework that transfers systematic thinking patterns from teacher models to smaller student models via **adaptive thinking-**

**pattern distillation**. It decomposes tasks into four staged modules—Problem Analysis, Logical Reasoning, Strategy Planning, and Retrieval Guidance—and emits concise, auditable plans that specify whether to retrieve, what to retrieve, and how to integrate evidence. To reduce retrieval overhead, the framework supports three composable paradigms (Figure 4): Forward Stepwise Reasoning, Backward Constraint Focusing, and Parallel Intersection Reasoning. Overall, FutureMind enables resource-constrained models to perform structured, low-latency retrieval and reasoning with improved accuracy and interpretability.

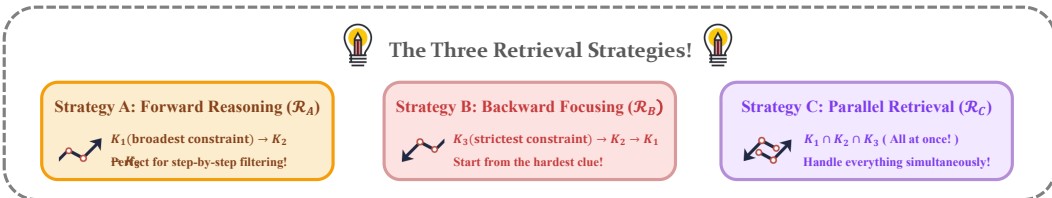

Figure 4: Three adaptive retrieval paradigms employed in FutureMind.

## E.2 Instruction for Enhanced ToolCall Using Parallel Search Tool C.1

> **Instruction for Enhanced ToolCall Using Parallel Search Tool**
>
> **System:**
> You are a helpful assistant.
>
> **Tools:**
> You may call one or more functions to assist with the user query. You are provided with function signatures within <tools></tools> XML tags:
>
> ```
> <tools>
>         {
>             "type": "function",
>             "function":
>                 {
>                     "name": "google_search",
>                     "description": "Parallel Search Tool of
>                         description",
>                     "parameters": "Parallel Search Tool of
>                         parameters",
>                 }
>         }
> </tools>}
> ```
>
> For each function call, return a json object with function name and arguments within <tools></tools> XML tags:
>
> ```
> <tool_call>
> {"name": <function-name>, "arguments": <args-json-object>}
> </tool_call>
> ```
>
> **User:**
> **Question: {user question}**
> You FIRST think about the reasoning process as an internal monologue and then provide the final answer. The reasoning process MUST BE enclosed within <think> </think> tags. The final answer MUST BE put in <answer> </answer> tags.

## E.3 INSTRUCTION FOR TOOLCALL-DRIVEN FUTUREMIND USING PARALLEL SEARCH TOOL C.1

---

**Instruction for ToolCall-driven FutureMind Using Parallel Search Tool**

**System:**
You are a helpful assistant.

**Tools:**
You may call one or more functions to assist with the user query.
You are provided with function signatures within <tools></tools> XML tags:

```
<tools>
        {
            "type": "function",
            "function":
                {
                    "name": "Parallel Search",
                    "description": "Parallel Search Tool of
                        description",
                    "parameters": "Parallel Search Tool of
                        parameters",
                }
        }
</tools>}

<tools>
        {
            "type": "function",
            "function":
                {
                    "name": "futuremind",
                    "description": "Futuremind Tool of
                        description",
                    "parameters": "Futuremind Tool of parameters",
                }
        }
</tools>}
```

For each function call, return a json object with function name and arguments within <tools></tools> XML tags:

```
<tool_call>
    {
        "name": <function-name>,
        "arguments": <args-json-object>
    }
</tool_call>
```

**user:**
**Question: {user question}**
You FIRST think about the reasoning process as an internal monologue and then provide the final answer. The reasoning process MUST BE enclosed within <think> </think> tags. The final answer MUST BE put in <answer> </answer> tags. First, invoke the "futuremind" tool to obtain a systematic thinking strategy and solution roadmap for the given Question (original question text verbatim). After that, you may call the search tool multiple times as needed to gather or verify information until you have sufficient material to answer. Once all necessary information is confirmed, provide the final answer using concise, focused language without unnecessary elaboration.

---

### E.4 RELATED METHODS: ENHANCED TOOLCALL (TC) AND TOOLCALL-DRIVEN FUTUREMIND (TC+FM)

**Enhanced ToolCall (TC).** Enhanced ToolCall (TC) is implemented on top of the ReAct framework, following the Toolcall Template D. To support efficient information access, we employ the **Parallel Search Tool** C.1, which enables parallel retrieval while constraining the reasoning process to a maximum of 10 steps. This design ensures both efficiency and controllability. The reasoning process follows the predefined instructions provided in E.2.

**ToolCall-driven FutureMind (TC+FM).** ToolCall-driven FutureMind (TC+FM) extends this design by incorporating **FutureMind (FM) Method** into the enhanced ToolCall framework. For more efficient and robust aggregation of retrieved evidence, we employ the **Parallel Search Tool** C.1 and **FutureMindTool** C.2, which enables parallel retrieval while constraining the reasoning process to a maximum of 10 steps. This design further improve retrieval robustness while maintaining controllable reasoning. The reasoning process follows the predefined instructions provided in E.3.

### E.5 MODULE OF FUTUREMIND: PROBLEM ANALYSIS-LOGICAL REASONING-STRATEGY PLANNING-RETRIEVAL GUIDANCE

In this section, we first introduce the instructions utilized by each module, followed by a detailed walkthrough using a concrete example. Specifically, we employ the query "A painter whose father died of heart disease, who has an elder sister, and who had five children with his wife before their marriage broke down. Later, he experienced three subsequent relationships. A literary work was later written based on this person. What is the title of that work?" to illustrate the step-by-step execution of each component within the FutureMind framework.

#### E.5.1 PROBLEM ANALYSIS MODULE

---
**Problem Analysis Module Instructions**

**Name of Module**: Problem Analysis
**Instruction**: Please first identify the core conflict, intrinsic attributes, and target outcomes of the problem; extract key dimensions $C_1$, $C_2$, $C_3$, $C_4$, ... (e.g., causal relationships, constraints, objective functions).

---
**Problem Analysis Module Execution Example**

**1. Core Objectives**
Precisely identify the painter who satisfies the specified biographical constraints and report the single literary work (title and author) that is documented as based on that painter, together with concise supporting evidence.

**2. Intrinsic Attributes**
Verifiable, atomic facts to check: (a) profession = painter; (b) father's cause of death = heart disease; (c) exactly one sister; (d) five children with his wife; (e) the marriage later dissolved; (f) subsequently experienced three distinct significant romantic relationships; (g) existence of a notable literary work explicitly linked to this person.

**3. Target Outcomes**
Output (1) the literary work's title and author; (2) a compact evidence map ( 3 bullet points) showing which of the above intrinsic attributes (by label) are corroborated by authoritative sources linking the painter to the work.

**4. Critical Conditions**
Let the four critical conditions be defined as:

- $C_1$ (**Identity / Profession**): "Painter" — restricts the candidate set to artists/painters.

---

- $C_2$ **(Family / Life Events):** Father died of heart disease; has one sister; had five children with his wife; the marriage later dissolved. These are distinctive and searchable biographical facts.

- $C_3$ **(Romantic History):** Later experienced three significant romantic relationships (multiple partners), with possible implications of relocation or cross-cultural life (common in comparable cases, though not explicitly stated).

- $C_4$ **(Literary Adaptation Clue):** A literary work was written based on this person — the target is the work's title (and its author), and the work is presumably well-known.

### E.5.2  LOGICAL REASONING MODULE

**Logical Reasoning Module Instructions**

**Name of Module**: Logical Reasoning
**Instruction**: Apply first principles to reverse-engineer the underlying logic, determine critical conditional elements (denoted $K_1$, $K_2$, $K_3$, $K_4$), and order them by importance or logical sequence.

**Logical Reasoning Module Execution Example**

Use first-principles decomposition to organize the constraints and decide the most efficient solving sequence.

- $K_1$: Had five children with his wife and the marriage ultimately broke down. Very specific life events that exclude most painters.

- $K_2$: Subsequently had multiple (approximately three) significant romantic relationships. Together with $K_1$ this narrows the candidate set substantially.

- $K_3$: Father died of heart disease (additional temporal/place details increase discriminative power). Serves as auxiliary confirmation.

- $K_4$: Explicitly identified as the prototype or clear inspiration for a notable literary work — provides the final verification link.

First search for individuals satisfying $K_1$; verify $K_2$ and $K_3$ in parallel; finally confirm the linkage to a notable literary work ($K_4$) to identify the unique match.

### E.5.3  STRATEGY PLANNING MODULE

**Strategy Planning Module Instructions**

**Name of Module**: Strategy Planning
**Instruction**: Thinking Strategies (Select one or more as appropriate).

**Strategy A: Forward Stepwise Reasoning (Progressive Filtering)**
• Sequentially filter from basic to decisive conditions:
a. Retrieve the set A meeting base condition $K_1$;
b. From A, filter the subset B meeting secondary condition $K_2$;
c. From B, filter the subset C meeting key condition $K_3$;
d. Within C, verify candidates satisfying decisive condition $K_4$.

**Strategy B: Backward Constraint Focusing (Reverse Narrowing)**
• Reverse-derive from stringent constraints to base compatibility:
a. Prioritize retrieving set A satisfying the strictest constraint (e.g., $K_3$) to narrow scope quickly;

b. From A, filter set B meeting critical feature condition (e.g., $K_4$);
c. From B, verify set C satisfying prerequisite condition (e.g., $K_2$);
d. Within C, confirm final candidates compatible with baseline condition (e.g., $K_1$).

**Strategy C: Parallel Intersection Reasoning (Parallel Filtering)**
• Treat conditions as independent dimensions in parallel:
a. Retrieve sets A, B, C, D each satisfying conditions $K_1$, $K_2$, $K_3$, $K_4$ respectively;
b. Compute intersection $(A \cap B \cap C \cap D)$ to extract solutions meeting all conditions simultaneously.

**Choose the strategy (or combination) based on problem characteristics, data availability, and efficiency requirements.**

---

Strategy Planning Module Execution Example

**Selected Strategy: Strategy B — Backward Constraint Focusing (recommended)**
**Rationale**: The "literary adaptation" condition ($K_4$) is the strongest and most selective constraint. Since only a small number of painters are explicitly used as prototypes for notable literary works, starting from this constraint yields the fastest and most discriminative narrowing of the search space.

**Strategy B: Backward Constraint Focusing (Reverse Narrowing)**
• Reverse-derive from the most stringent constraint ($K_4$):
a. Begin with $K_4$: retrieve or enumerate well-known literary works explicitly based on, or inspired by, real painters.
b. For each identified literary work, extract the painters documented or claimed as prototypes.
c. For each candidate painter, verify biographical compatibility with $K_1$ (profession), $K_2$ (family/life events), and $K_3$ (romantic history).
d. Remove any candidate failing one or more conditions; if no candidate remains, return to step (a) and expand the set of literary works.
e. The painter who satisfies $K_1$–$K_3$ and matches $K_4$ constitutes the final solution.

Apply Strategy B exclusively. Start from the strongest constraint (literary adaptation), identify candidate works and their prototypes, then filter by $K_1$–$K_3$ to obtain the unique match.

---

### E.5.4 RETRIEVAL GUIDANCE MODULE

Retrieval Guidance Module Instructions

**Name of Module**: Retrieval Guidance
**Instruction**: Thinking Strategies (Select one or more as appropriate).

**Keyword Guidance**
Derive core entities, attributes, and relationships from the problem analysis (e.g., "AI + ethical risks + regulatory policy").

**Resource Guidance**
List authoritative sources relevant to the domain: academic databases, industry reports, government or standards bodies, etc. (e.g., PubMed, IEEE Xplore, World Bank Data).

**Sequence Guidance**
Design the retrieval sequence by priority or logic (e.g., first industry standards, then empirical studies, finally policy documents).

> **Query Guidance**
> Formulate layered/combined search expressions supporting iterative filtering:
> e.g., ("AI ethical risks" AND "regulatory policy") OR ("autonomous driving safety standards" AND "EU regulations"), or retrieve with keyword A first, then refine results with keyword B.
>
> **Screening Guidance**
> Define preliminary inclusion/exclusion criteria and relevance assessment to support deeper analysis later.

---

**Retrieval Guidance Module Execution Example**

**1. Keyword Design (Hierarchical)**

- Level 1 (Directly locate works/prototypes): "novel based on painter", "fiction inspired by painter"
- Level 2 (Combine with biographical details): "painter five children divorced wife", "father died of heart disease painter", "painter with sister"
- Level 3 (Verify person and work combinations): "<Work Title> prototype", "<Painter Name> depicted as <Work Title>"

**2. Resource Selection (Priority)**

- Authoritative biographies and academic publications (artist biographies, art history books)
- Literary studies and annotations (research articles on novel prototypes, author memoirs)
- Recognized encyclopedias and databases (e.g., Wikipedia for preliminary screening, with professional biographies as final reference)
- Cultural and historical journals, author studies, and book reviews (to verify claims of works based on real persons)

**3. Retrieval Sequence (Recommended)**

- First retrieve well-known novels or works based on painters to generate candidate works meeting the key literary adaptation condition
- Examine annotations, author notes, and academic reviews of these works to identify the associated prototype painters
- Retrieve biographies of candidate painters to verify biographical details such as number of children, marital status, father's cause of death, and romantic history
- Cross-validate with independent authoritative sources for final confirmation

**4. Query Examples (Facilitating Iterative Filtering)**

- "novel inspired by painter" OR "fiction based on painter"
- "painter AND five children AND divorced wife"
- "'<Work Title>' prototype <Painter Name>" (used to verify known candidates)

**5. Screening Criteria (Preliminary Inclusion / Exclusion)**

- Inclusion: Reliable sources such as biographies, academic papers, authoritative book reviews, and author statements; multiple independent sources citing the same prototype and facts
- Exclusion: Unverified blogs or secondhand claims without sources; circular references lacking original evidence; candidates with biographical details contradicting key facts (e.g., children count, marital status)

# F  INSTRUCTION TEMPLATES

## F.1  INSTRUCTIONS FOR EVALUATION

In this work, we use LLM-as-Judges (GPT-4o-mini) to evaluate multi-hop question answering (QA) benchmarks: 2WikiMultihopQA, Bamboogle, MuSiQue, and FRAMES. The specific instructions are as follows.

---

**Instruction for Judge**

Given a Question and its Golden Answer, verify whether the Predicted Answer is correct. The prediction is correct if it fully aligns with the meaning and key information of the Golden Answer. Respond with True if the prediction is correct and False otherwise.

[Question:] **{user question}**
[Golden Answer] **{reference answer}**
[Predicted Answer] **{assistant's answer}**

---

## F.2  INSTRUCTION FOR NAIVE GENERATION

---

**Instruction for Naive Generation**

Please answer the below questions.You should think step by step to solve it.The final answer MUST BE put in <answer> </answer> tags.

[Question:] **{user question}**

---

## F.3  INSTRUCTION FOR STANDARD RAG

---

**Instruction for Standard RAG**

You are a knowledgeable assistant that utilizes the provided documents to answer the user's question accurately.

Guidelines:
- Analyze the provided documents to extract relevant information. Synthesize the information to formulate a coherent and accurate answer.
- Ensure that your response directly addresses the user's question using the information from the documents.

[Question:] **{user question}**
[Documents:] **{documents}**

---

## F.4  INSTRUCTION FOR SEARCH-O1

---

**Instruction for Search-o1**

You are a reasoning assistant with the ability to perform web searches to help you answer the user's question accurately. You have special tools:
To perform a search: write <|begin_search_query|> your query here <|end_search_query|>.
Then, the system will search and analyze relevant web pages, then provide you with helpful information in the format <|begin_search_result|> ...search results... <|end_search_result|>.
You can repeat the search process multiple times if necessary. The maximum number of search attempts is limited to {MAX_SEARCH_LIMIT}.
Once you have all the information you need, continue your reasoning.

---

> Example:
> Question: "..."
> Assistant thinking steps:
> - I might need to look up details about ...
> Assistant:
> <|begin_search_query|>...<|end_search_query|>
> (System returns processed information from relevant web pages)
> Assistant continues reasoning with the new information...
> Remember:
> - Use <|begin_search_query|> to request a web search and end with <|end_search_query|>.
> - When done searching, continue your reasoning.

# G  FORMULA DESCRIPTION

## G.1  FORMULA EXPLANATION OF $\text{ACC}_E$

**Formula:**

$$\text{ACC}_E = \begin{cases} 1 & \exists g \in G, \text{norm}(pred) \supseteq \text{norm}(g) \\ 0 & \text{otherwise} \end{cases}$$

where $pred$ is the predicted answer; $G = \{g_1, \ldots, g_k\}$ is the gold answer set; $\text{norm}(\cdot)$ denotes a normalization procedure that handles paraphrasing, punctuation, and related surface variations; and the relation $A \supseteq B$ indicates that $A$ fully covers the core semantics of $B$. Hence, $\text{ACC}_E = 1$ when the normalized prediction semantically covers at least one normalized gold answer, and $0$ otherwise.

# H  OVERALL TOKEN CONSUMPTION AND API COST OF GPT-4O-MINI AND GOOGLE SEARCH API

## H.1  COST ANALYSIS OF LARGE MODEL EVALUATION

For the GPT-4o-mini model, the total number of tokens consumed during the scoring process can be divided into three main components: the initial scoring prompt F.1, the question, the golden predicted answer, and the GPT-4o-mini model's output (True/False).

We statistically analyzed the input token usage across different datasets, as shown in Table 8. Across the four datasets (2WikiMQA, Bamboogle, Frames, and MuSiQue), the total input token count amounts to approximately 772,000 tokens, while the total output token count is around 2,000 tokens.

Given the official pricing of GPT-4o-mini — $0.15 per million input tokens and $0.60 per million output tokens — the total API cost for one complete experimental setting across all datasets is estimated to be approximately $0.117. In total, we conducted 84 such experimental runs, resulting in an overall evaluation cost of approximately $9.828 for the GPT-4o-mini scoring process.

Table 8: Overall token consumption of GPT-4o-mini across different datasets in a single experimental setting. All values are reported in units of *10,000 tokens (w)*.

| Dataset | Tokens Consumed (w) |
| --- | --- |
| 2WikiMQA | 16.5 w |
| Bamboogle | 3.7 w |
| Frames | 38.0 w |
| MuSiQue | 19.0 w |
| **Total** | **77.2 w** |

Table 9: Total number of search queries across datasets for each teacher–student model pair under Baseline and FutureMind settings.

| Dataset | Qwen-72B $\rightarrow$ 3B | | Qwen-32B $\rightarrow$ 3B | | Qwen-14B $\rightarrow$ 3B | | Qwen-7B $\rightarrow$ 3B | |
|---|---|---|---|---|---|---|---|---|
| | TC | TC+FM | TC | TC+FM | TC | TC+FM | TC | TC+FM |
| Wiki | 1032 | 1297 | 1028 | 1120 | 987 | 1322 | 1005 | 1259 |
| Bamboogle | 208 | 235 | 203 | 184 | 196 | 286 | 202 | 214 |
| Frames | 2030 | 1923 | 2000 | 1210 | 1693 | 2031 | 2016 | 1492 |
| Musique | 1056 | 1217 | 1032 | 821 | 1021 | 821 | 1040 | 997 |
| Total | 4326 | 4672 | 4263 | 3335 | 3897 | 4460 | 4263 | 3962 |

## H.2 COST ANALYSIS OF SEARCH TOOL INVOCATION

This cost estimation focuses exclusively on the direct expenses incurred by the *Enhanced ToolCall* method and the *ToolCall-driven FutureMind* method. For both approaches, the majority of the cost originates from API calls made by the *parallel Search Tool*. According to the official pricing of the Custom Google Search JSON API, the cost is set at $5 per 1,000 queries.

Across the four datasets (*2WikiMQA*, *Bamboogle*, *Frames*, and *MuSiQue*), both the Enhanced ToolCall and ToolCall-driven FutureMind methods perform approximately 4.2k search engine queries on average. Therefore, the estimated cost for executing all experiments across the four datasets with either method is roughly $21. Considering the total number of experimental runs, the cumulative search-related expenditure is estimated to be approximately $1,365.

In addition, we provide a concrete example to illustrate this estimation. In this case, the student model is *Qwen2.5-3B*, and the teacher models are *Qwen2.5-72B*, *Qwen2.5-32B*, *Qwen2.5-14B*, and *Qwen2.5-7B*, respectively. The detailed statistics are presented in Table 9, which reports the total number of search queries across datasets for each teacher–student model pair under both the *Enhanced ToolCall*(TC) and *ToolCall-driven FutureMind*(TC+FM) settings.

## I  DISCUSSION: FUTUREMIND'S FLEXIBILITY IN CORRECTING THE LOGICAL PATH

This section discusses how **FutureMind** exhibits flexibility in correcting and refining the logical reasoning path. This capability represents one of the core design goals of FutureMind: to enable dynamically adaptive reasoning that balances structured planning with real-time flexibility.

Rather than functioning as a one-shot, fixed-plan generator, FutureMind is designed as a **dynamically callable tool** (clarified in Appendix E.3) that can be invoked iteratively as needed. This design preserves structured planning capabilities while maintaining flexibility in reasoning, thereby enabling more adaptive inference strategies. Its core mechanisms include:

- On-demand triggering. FutureMind can be called on demand when the student model detects issues such as an excessively large candidate space, high retrieval noise, or invalid intermediate states. In these situations, FutureMind provides refined or alternative reasoning strategies to guide subsequent exploration.

- Strategy diversity with decentralized control. FutureMind provides dynamic and diverse reasoning strategies, while the ultimate selection and adjustment of the reasoning path are made by the student model based on real-time retrieval results and available computational budget, preserving flexibility.

Overall, this interaction forms a closed adaptive loop of **execution bottleneck** $\rightarrow$ **on-demand invocation** $\rightarrow$ **strategy optimization**, mirroring the iterative problem-solving logic of human experts.

**Illustrative Example.** The following case demonstrates this mechanism through a knowledge-intensive reasoning example: Who is the poet that was a friend of the author of One Hundred Years of Solitude and also won a Nobel Prize?

- **First FutureMind call.** During the initial exploration, the model recognized that retrieving "friends" could lead to a large number of redundant candidates. It therefore made the first FutureMind call and obtained a **forward stepwise reasoning strategy**: Identify the author → Retrieve well-known literary associates → Filter for poets → Verify Nobel laureateship.

- **Second FutureMind call.** During execution, even after narrowing the set of "associates", dozens of candidates remained, indicating high downstream reasoning difficulty. The model triggered a second FutureMind call and received a recommended **backward constraint–focusing strategy**: List Nobel Prize–winning poets → Filter candidates with documented associations to García Márquez → Confirm friendship via credible sources.

- After receiving this guidance, the model refined the second step to "filter candidates with publicly verifiable interaction records with García Márquez" and proceeded. It first identified **Pablo Neruda** (1971 Nobel Prize in Literature, poet) as the core candidate and confirmed their long-term friendship via Márquez's essay **García Márquez on Pablo Neruda**. Verification through literary archives ultimately yielded the correct answer **Neruda**.

## J    CASE STUDY

We present a case study to illustrate the full pipeline of the proposed FutureMind method 2. The case is drawn from the MuSiQue benchmark, which poses a question that the Qwen2.5-7B-Instruct model alone can hardly answer correctly. We then demonstrate how the proposed ToolCall-driven FutureMind employs Qwen2.5-14B-Instruct as a teacher model to guide Qwen2.5-7B-Instruct in solving the problem, and provide the full reasoning trajectory leading to the final answer.

---

**Case Trajectory in MuSiQue**

**Question:** When was the Palau de la Generalitat built in the city in which died the man called Martin of the region of Spain where Berdejo is located?

**Step 1**

**\<think\>**
I'm not fully familiar with this question, so I'll first call the FutureMind tool to get a systematic thinking pattern and roadmap.
**\</think\>**

**\<tool_call\>**
{"name": "FutureMind", "arguments": {"query": ["When was the Palau de la Generalitat built in the city in which died the man called Martin of the region of Spain where Berdejo is located?"]}}
**\</tool_call\>**

**\<tool_response\>**
1. **Problem Analysis** - Identify the construction date of the Palau de la Generalitat in the city where Martin of the region of Spain, where Berdejo is located, died.

2. **Logical Reasoning** - Key conditional elements
      K1 (correct identification of Martin and his region in Spain)
      K2 (connection between Martin and the city of death)
      K3 (location of Berdejo in Spain)
      K4 (construction date of the Palau de la Generalitat in the identified city)

3. **Strategy Planning** - Selected strategy A (Forward Stepwise Reasoning)
      a. Retrieve information on Martin of the region in Spain where Berdejo is located (K1)
      b. From the information, identify the city where Martin died (K2)

---

    c. Verify the connection between Berdejo and the identified region in Spain (K3)

    d. Within the identified city, find the construction date of the Palau de la Generalitat (K4)

4. **Retrieval Guidance**

    **Keywords:** Martin of [region in Spain], Berdejo, Palau de la Generalitat, construction date

    **Resources:** Wikipedia, Encyclopedia Britannica, Spanish historical archives, local government websites of the identified region and city

    **Sequence:** Historical biographies → regional information → local historical records → architectural or government websites

    **Query:** "Martin of [region in Spain] biography"; "Berdejo location in Spain"; "Palau de la Generalitat construction date" in the identified city

    **Screening:** Exclude non-relevant biographies or historical records; limit to sources with verifiable information on the construction date.

**</tool_response>**

---

**Step 2**

**<think>**
To answer the question, the goal is to identify the construction date of the Palau de la Generalitat in the city where Martin, from the region in Spain where Berdejo is located, died. The key steps are: 1. First, gather information about Martin and the region where Berdejo is located. This will allow the identification of the city where Martin died. 2. Then, validate the connection between Berdejo's location and the identified region in Spain. 3. After identifying the city where Martin died, find the construction date of the Palau de la Generalitat in that city. The selected strategy is Forward Stepwise Reasoning, and the reasoning involves retrieving historical and regional data followed by architectural information for the Palau de la Generalitat. The next step is to search for relevant data using the following queries.
**</think>**

**<tool_call>**
{"name": "google_search", "arguments": {"query": ["Martin of [region in Spain] biography", "Berdejo location in Spain", "Palau de la Generalitat construction date"]}}
**</tool_call>**

**<tool_response>**
Martin (born 1356, Gerona, Catalonia [Spain]—died May 31, 1410, Barcelona) was the king of Aragon from 1395 and of Sicily (as Martin II from 1409) (https://www.britannica.com/biography/Martin-king-of-Aragon-and-Sicily)
From March 2 to 14, 2024, the Palau de la Generalitat de Valencia opens its doors and invites you to explore this historic building that houses the Presidency (https://www.visitvalencia.com/en/events-valencia/palau-de-la-generalitat-opens-its-doors-celebrate-fallas)
...
**</tool_response>**

---

**Step 3**

**<think>**
The search results reveal the following: 1. Several individuals named Martin exist in different regions of Spain, but Martin I of Aragon (1392–1409) seems the most relevant. However, further information about his death location is needed. 2. Berdejo is a town in the Zaragoza province of Aragon, Spain. 3. The Palau de la Generalitat in Barcelona dates back to the 15th century. Given the lack of a direct link between Martin I and Berdejo, we will refine the search to focus on identifying the city where Martin I died, and its potential connection to Berdejo. The next search query will be: Martin I of Aragon death place.
**</think>**

**<tool_call>**
{"name": "google_search", "arguments": {"query": ["Martin I of Aragon death place"]}}
**</tool_call>**

**<tool_response>**
Matino is the son of King Martin I of Aragon. In the 1390, Matino married the Sicilian Queen and cousin Maria and became co-king. After Matino's death in 1409, his father became King of Sicily, known as Matino (https://www.wikiwand.com/zh-cn/articles)
...
**</tool_response>**

| Step 4 |
| :---: |

**<think>**
Martin I of Aragon died in Barcelona on May 31, 1410. The question asks for the construction date of the Palau de la Generalitat in Barcelona, where Martin died, and relates it to Berdejo, located in Zaragoza, Aragon. Since Berdejo is in a different region (Aragon) and the Palau de la Generalitat is in Barcelona (Catalonia), we need to focus on the construction of the Palau de la Generalitat in Barcelona. This was completed in the 15th century, and since Martin I died in 1410, the building was already constructed by then.
**</think>**

**<answer>**The Palau de la Generalitat in Barcelona, Spain, was built in the 15th century, long before Martin I of Aragon died in 1410. **<answer>** ✓

