# OpenReview forum: "FutureMind: Equipping Small Language Models with Strategic Thinking-Pattern Priors via Adaptive Knowledge Distillation"
_ICLR.cc/2026/Conference — ICLR 2026 Poster_

### Official Review · Reviewer_xqBy · 2025-10-27

**Soundness:** 3
**Presentation:** 3
**Contribution:** 3
**Rating:** 6
**Confidence:** 4

**Summary:**

This paper proposes FutureMind, a training-free four-stage framework with Problem Analysis, Logical Reasoning, Strategy Planning, and Retrieval Guidance. Experimental results demonstrate the effectiveness of the proposed method on four multiple-hop question answering datasets.

**Strengths:**

1. The proposed method is intuitive and well-elaborated.
2. The proposed method demonstrates effectiveness and generalizability on four multi-hop QA datasets, using LLMs of different series and sizes.

**Weaknesses:**

1. This paper claims to be "a new state of the art among training-free methods", but only Naive Generation, Standard RAG, and Search-o1 are compared, ignoring various baselines on inference-time scaling methods and training-free LLM frameworks.
2. Section 4.1 Datasets: "we randomly sample 500 instances from the validation sets of 2WikiMQA and MuSiQue". However, 2Wiki has about 12.6k validation instances, and MuSiQue has 2.4k validation instances. Sampling only 500 per dataset here is too limited.
3. Section 4.4 Ablation Studies: This part only shows the ablation of different strategies in the "Strategy Planning" module. However, it is even more important to investigate the ablation of each module/stage, i.e., Problem Analysis, Logical Reasoning, Strategy Planning, and Retrieval Guidance.
4. The proposed method utilizes more inference-time computing, but an efficiency study is lacking.

**Questions:**

**Questions**:
1. Section 4.1 Metrics: What exactly is the formula of $ACC_{E}$?

**Suggestions**:
1. It is suggested to mention the reason for naming the proposed method as "FutureMind" in the paper.
2. It is suggested to include the costs of calling the LLM judge and Google Search API.
3. Line 84: "training-free" or "inference-only" sounds better to me, as "zero-training-cost" can be ambiguous: it could be a process without training OR a process with some zero-cost "training".
4. Line 467 "As shown in Table 1 and Table 4": As Table 1 is on Page 7 and Table 4 is on Page 9, it would be better to present the performance changes ($\Delta$ ACC) in Table 4, or illustrate the differences using bar charts.

---

> ### Author Response · Authors · 2025-11-20
> **1. Baseline Selection and Explanation of "New State of the Art" Claim.**
>
> &emsp; We thank the reviewer for the comment. The proposed modular reasoning framework, FutureMind, primarily targets training-free reasoning scenarios, where the core mechanism is to dynamically acquire thinking-pattern priors from a teacher model to support more efficient strategy planning and retrieval. To ensure a fair evaluation, we selected three widely used training-free baselines: **Naive Generation**, **Standard RAG**, and **Search-o1**, representing direct generation, retrieval-augmented generation, and self-initiated retrieval generation, respectively. Among them, Search-o1 is regarded as one of the strongest training-free frameworks in the domain.
>
> &emsp; Experimental results demonstrate that, under identical training-free settings, FutureMind consistently yields substantial improvements across different architectures and scales of small language models. Therefore, we describe it as achieving "new state of the art" among training-free methods, strictly within the scope of this experiment setting.

---

> ### Author Response · Authors · 2025-11-20
> **2. Discussion on Inference-Time Computing.**
>
> &emsp; We thank the reviewer for pointing out this aspect. FutureMind indeed introduces additional inference-time latency, primarily due to the strategy-generation overhead from teacher models. However, we argue that the core metric for evaluating reasoning efficiency is not the per-step latency but rather **the number of retrieval tool invocations**.
>
> &emsp; In complex tasks, FutureMind substantially reduces unnecessary retrieval calls, thereby constructing **shorter and more efficient solution paths** and avoiding large cumulative costs from "blind retrieval" in the model’s reasoning process. For example, on the Frames dataset, the baseline model required 1,286 Google Search API calls, whereas FutureMind invoked only 649 API calls — **a reduction of 637**. This demonstrates that even with added latency from teacher-model reasoning, FutureMind achieves a more efficient overall solution path and lower retrieval cost.

---

> ### Author Response · Authors · 2025-11-20
> **3. Dataset Sampling Methodology.**
>
> &emsp; We thank the reviewer’s question about sampling. Our study involves over **300 reasoning experiments** across 6 student models of different sizes and architectures, 4 datasets of varying difficulty, and multiple settings. To ensure feasibility and control costs, we randomly sampled 500 instances from each dataset, while maintaining the reproducibility of our experiments.
>
> &emsp; In future work, we plan to perform experiments across different datasets using consistent sampling ratios to further improve robustness.

---

> ### Author Response · Authors · 2025-11-20
> **4. Ablation Study Explanation.**
>
> &emsp; FutureMind contains four **serial** modules: **Problem Analysis**, **Logical Reasoning**, **Strategy Planning**, and **Retrieval Guidance**. The Problem Analysis module establishes the model’s understanding of the question by decomposing it into structured elements. The Logical Reasoning module then derives key conditions based on these structured elements, taking into account logical dependencies and their relative importance, guided by first-principles reasoning. Within the Strategy Planning module, we designed three adaptive retrieval paradigms (forward stepwise reasoning, backward constraint focusing, and parallel intersection reasoning) to efficiently integrate evidence for complex problems.
>
> &emsp; As FutureMind is strictly **sequential**, removing any preceding module would severely disrupt the execution of subsequent modules. Therefore, our ablation focuses on evaluating the three adaptive paradigms within the Strategy Planning module. Ablations for the other modules have been added in Table 3 of the revised manuscript for completeness.

---

> ### Author Response · Authors · 2025-11-20
> **5. Clarification of Metric Formula.**
>
> &emsp; We thank the reviewer for noting the need to clarify the evaluation metric. In this work, we adopt an exactness-oriented semantic matching metric, denoted as \\(\\mathrm{ACC}_E\\), which evaluates whether the model’s predicted answer semantically covers at least one element of the gold answer set.
>
> &emsp; Specifically, the metric is defined as:
>
> \\[
> \\mathrm{ACC}_E \\;=\\;
> \\begin{cases}
> 1, & \\exists\\, g\\in G\\ ,  \\ \\operatorname{norm}(\\mathrm{pred}) \\supseteq \\operatorname{norm}(g),\\\\[6pt]
> 0, & \\text{otherwise.}
> \\end{cases}
> \\]
> where \\(\\mathrm{pred}\\) is the predicted answer; \\(G=\\{g_1,\\dots,g_k\\}\\) is the gold answer set; \\(\\operatorname{norm}(\\cdot)\\) denotes a normalization procedure that handles paraphrasing, punctuation, and related surface variations; and \\(A\\supseteq B\\) indicates that \\(A\\) fully covers the core semantics of \\(B\\).
>
> &emsp; Hence, \\(\\mathrm{ACC}_E=1\\) when the normalized prediction semantically covers at least one normalized gold answer, and \\(\\mathrm{0}\\) otherwise.
>
> &emsp; This revision provides a clearer explanation.

---

> ### Author Response · Authors · 2025-11-20
> **6. Suggestions for Paper Revisions.**
>
> We thank the reviewer for the constructive suggestions.
> - The name "FutureMind" is intended to reflect a vision for **future** AI systems that, like humans, can flexibly apply structured reasoning strategies. The model leverages **thinking-pattern priors** without relying on training data or fine-tuning, allowing it to generalize to high-difficulty, unseen problems. We will clarify this in the revised manuscript.
> - We will include a cost analysis for invoking the **LLM judge** and the **Google Search API** in Appendix~I of the revised manuscript. Across all experimental runs, the GPT-4o-mini scoring process incurred a total cost of approximately \\$9.828， the Google Search API cost amounts to approximately \\$1365.
> - We agree that "training-free" is more precise than "zero-training-cost" and will adopt this terminology in the revised manuscript.
> - Regarding the presentation of performance changes, we have redrawn the bar charts (Figure 3) to more intuitively and clearly illustrate the performance differences, facilitating a better understanding of the model improvements by readers. The updated visualizations have been incorporated into the revised manuscript as a valuable complement to the data in Tables 1 and 4.

---

> ### Comment · Reviewer_xqBy · 2025-11-28
> **Rebuttal Acknowledgement by Reviewer xqBy**
>
> Thanks for the detailed replies.
>
> - I appreciate the explanations for the proposed Weaknesses 1, 2, and 3, while I still keep the concerns.
> - Please include the discussion about W4 in the revision.
> - The new Figure 3 looks good, and it can be clearer if the font size is larger.
>
> Overall, I maintain a positive rating of the submission.

---

> > ### Author Response · Authors · 2025-11-28
> >
> > &emsp;Thank you very much for your constructive feedback and for maintaining a positive rating of our submission.
> >
> > &emsp;We sincerely appreciate your recognition of our work. The additional modules ablation analyses and the improved version of Figure 3 have already been updated in the revised manuscript. Moreover, your concerns regarding reasoning latency are well noted, and we will pay special attention to this aspect in our future work.
> >
> > &emsp;**Wish you a pleasant day!**

---

### Official Review · Reviewer_t5P7 · 2025-10-30

**Soundness:** 2
**Presentation:** 2
**Contribution:** 3
**Rating:** 4
**Confidence:** 3

**Summary:**

The authors present **FutureMind**, a training-free modular reasoning framework designed to enhance the reasoning and retrieval capabilities of small language models (SLMs). FutureMind leverages a large language model (LLM) as a planner that guides the SLM through multi-hop reasoning tasks. The framework comprises four modules: Problem Analysis, Logical Reasoning, Strategy Planning, and Retrieval Guidance. Through experiments on several multi-hop QA benchmarks, using models from the `Qwen2.5` and `Llama3.1` families, the authors demonstrate that FutureMind achieves state-of-the-art performance among training-free methods. They also find that naively increasing the size of the LLM planner does not necessarily yield better results, highlighting the importance of compatibility between planner and reasoning models.

**Strengths:**

- The main idea of the paper is interesting and has clear merit. Leveraging larger LMs to guide smaller ones during multi-hop reasoning is a natural and well-motivated direction. The method is also training free, something that showcases its efficiency and potential to be deployed.
- The experimental results are promising. Across all model scales, integrating FutureMind consistently outperforms the other baselines in almost all cases, which highlights its potential.
- The ablation study in Section 4.4 provides further evidence supporting FutureMind’s design, as removing any of the considered retrieval strategies from the planner leads to performance degradation across benchmarks.

**Weaknesses:**

- I think that the way each module is presented in Section 3, although detailed, is unnecessarily abstract and ends up confusing the reader instead of describing the modules clearly. Also, many parts are not adequately explained (such as what exactly is function $\mathcal{F}$), and there is a gap between the high-level overview of each module, and the actual implementation (for instance, it is not clear how the authors prompt the larger LM at each step of the pipeline).
	- A suggestion could be to move the instructions of Appendix E.5 to the main body of the paper, and move the more abstract definitions to the Appendices, (for completeness only).
- I am not sure whether I agree with the author's conclusions regarding the "cognitive bias bottleneck". I wouldn't describe FutureMind as a form of "knowledge distillation", nor as "lossy compression". I believe that the main observations of Table 2 can be attributed to the *capability gap* of the SLM and the planner LLM: the planner, being more capable compared to the reasoning SLM, generates a plan that overestimates the capabilities of the SLM. Thus, I am not sure whether one can claim that "noise is amplified during knowledge distillation".

**Questions:**

- Are the instructions of Appendix E.5 the exact prompts used?
- Could you clarify why the method is described as a form of knowledge distillation? From the paper, it appears that the large LM serves primarily as a planner for the smaller model, rather than transferring knowledge in the conventional sense.

---

> ### Author Response · Authors · 2025-11-20
> **1. Clarification of FutureMind as a "Knowledge Distillation" Method.**
>
> &emsp; We fully understand the reviewer’s concerns regarding the conventional definition of "knowledge distillation". In this paper, our use of the term does not refer to the traditional distillation paradigm centered on representation compression or soft‑label transfer. Instead, it aligns with the recent and expanding line of research on **reasoning distillation** [1][2].
>
> &emsp; Within this broader context, we view FutureMind as a reasoning distillation framework. Its goal is not to make the student model replicate the teacher’s factual knowledge, but rather to enable the student to selectively draw upon the teacher’s **thinking-pattern priors** according to its own capabilities and the demands of the current task, and incorporate these priors into its retrieval planning and reasoning process.
>
> &emsp; We will clarify this terminology and conceptual positioning in the revised manuscript to avoid confusion with the traditional definition of "knowledge distillation".
>
> ---
>
> **References**
>
> 1. Do, Cong Thanh, et al. _"Effectiveness of Chain-of-Thought in Distilling Reasoning Capability from Large Language Models."_ Proceedings of the 18th International Natural Language Generation Conference (2025).
> 2. Hsieh, Cheng‑Yu, et al. _"Distilling step-by-step! Outperforming larger language models with less training data and smaller model sizes."_ Findings of ACL (2023).

---

> ### Author Response · Authors · 2025-11-20
> **2. Explanation of the Role Orientation of the Teacher Model.**
>
> &emsp; Regarding the role orientation of the teacher model, we fully understand the reviewer’s concern. In the traditional sense, a planner typically focuses on providing explicit and executable action-level instructions. However, in our work, the teacher model does not prescribe concrete operational steps. Instead, through the four reasoning modules, it generates **thinking-pattern priors** that offer the student model a high-level framework of logical reasoning and principles for tool usage.
>
> &emsp; During inference, the student model is not a passive executor of the teacher’s plans. Rather, it selectively absorbs, references, or adapts these thinking patterns based on its own capabilities and the requirements of the current task, thereby improving its retrieval planning and reasoning performance. In this sense, the teacher model in FutureMind functions more as a provider of **structural reasoning at the strategy level**, rather than a planner issuing step-by-step commands.

---

> ### Author Response · Authors · 2025-11-20
> **3. Explanation of the "Cognitive Bias Bottleneck", "Lossy Compression" and "Noise Amplification".**
>
> &emsp; We thank the reviewer for the in-depth discussion and agree that these concepts require clearer articulation. Our experiments reveal a key phenomenon: **cognitive compatibility outweighs raw scale**. In other words, a larger teacher model does not necessarily provide greater benefits to the student model. We explain this observation in detail below.
>
> &emsp; Take Qwen-3B as the student model. Our results show that the **14B teacher outperforms the 32B teacher** (ACC_L: 29.82 > 24.25) in Table 2. If this pattern were explained solely by the student’s capacity ceiling, then the 32B teacher would produce strategies beyond the 3B student’s interpretability, and increasing teacher size further would not be expected to improve performance. However, we observe that the **72B teacher again outperforms the 14B teacher** (ACC_L: 30.41 > 29.82), contradicting the simple capacity-ceiling hypothesis. Thus, the non-monotonic relationship between teacher size and student performance cannot be attributed solely to student model capacity.
>
> &emsp; We attribute this phenomenon to what we describe as the "cognitive bias bottleneck". Concretely, a student model more readily internalizes strategy knowledge from teachers with a smaller cognitive gap — for example, those whose parameter distributions, training data, or optimization dynamics are more similar to the student’s. When the teacher’s reasoning structure becomes overly complex or its expression style diverges substantially from the student’s, the student tends to simplify the strategy during uptake and may drop crucial structural dependencies, resulting in performance degradation.
>
> &emsp; In the paper, we originally referred to this effect as "lossy compression", meaning the student cannot fully understand or apply the teacher’s strategy knowledge. The parts of the teacher’s output that cannot be effectively interpreted by the student become "noise", and such noise is amplified with increasing cognitive mismatch between teacher and student. We note, however, that the term "lossy compression" can be misleading. In the revised manuscript we will adopt the more precise term "strategic information loss" to describe the student’s loss of critical strategy information during interpretation.

---

> ### Author Response · Authors · 2025-11-20
> **4. Detailed Description of the FutureMind Method.**
>
> - As for Question 1, the instructions in Appendix E.5 are exactly the prompts used.
>
> - Function \\(\\mathrm{\mathcal{F}}\\) refers to the use of the teacher model to perform a preliminary evaluation of retrieval plans, in order to generate recommendations for the optimal retrieval strategy.
>
> - We sincerely thank the reviewer for the suggestion regarding the presentation of our method. We fully agree that a clearer depiction of module implementation details would improve the readability and reproducibility of the paper. In the current version, the abstract definitions in Section 3 are primarily intended to formalize FutureMind as a general reasoning framework applicable across different tasks. We also recognize that this level of abstraction may reduce the intuition of implementation details. To address this, we have further supplemented the module descriptions in Section 3 and indicated the implementation approaches of the four modules in the main text, while providing detailed execution examples in the appendix to facilitate reader understanding.
>
> These updates will be incorporated into the revised manuscript.

---

### Official Review · Reviewer_4F63 · 2025-11-06

**Soundness:** 2
**Presentation:** 3
**Contribution:** 3
**Rating:** 6
**Confidence:** 3

**Summary:**

The authors propose a framework named FutureMind, which enables SLMs to inherit the strategic planning capabilities of LLMs through a clever "thinking paradigm distillation." This allows SLMs to efficiently solve complex reasoning tasks without additional training. The work presents a unique perspective, and the experimental results are solid.

**Strengths:**

1. The most interesting aspect of this work is its redefinition of knowledge distillation. While traditional distillation often involves the student mimicking the teacher's trajectory, FutureMind empowers the student to master the teacher's structured thinking framework.

2. The experimental results reveal that an overly complex plan can become a barrier to the SLM's understanding and execution. This is a counter-intuitive and interesting phenomenon, for which the authors have provided a corresponding explanation.

3. The framework's design is exceptionally clear. It delegates the planning task to the more capable LLM while assigning the relatively simpler execution task to the efficient SLM, offering a plug-and-play solution.

**Weaknesses:**

1. I would like to ask the authors if they have considered or attempted to use more direct metrics to quantify or predict the "cognitive compatibility" between teacher and student models, beyond observing final performance in experiments. For instance, could the distillation effectiveness be anticipated by analyzing the relationship between the complexity of the teacher's plan and the capability limits of the student model?

2. The experiments in this paper are primarily focused on structured, multi-hop question-answering tasks, where the FutureMind's P-L-S-R process excels. I am interested in the authors' views on the framework's potential in more open-ended and creative tasks, such as code generation. In such tasks, the "critical conditions" and "logical sequence" of a problem may be less explicit. Would FutureMind's structured planning still be applicable, or would it require adjustments?

3. I have a concern that FutureMind makes a premature and irrevocable commitment to an entire logical path before any actual evidence is gathered. This contrasts with the exploratory and iterative methodology employed by LLMs in "thinking" mode, and indeed by human experts, when solving complex problems. For example, when faced with a problem that has a vast search space or high ambiguity, a seemingly reasonable initial plan could prove disastrous during execution. Consider the question: "Who is the poet that was a friend of the author of 'One Hundred Years of Solitude' and also won a Nobel Prize?"

    - A powerful teacher model would likely generate a seemingly perfect "forward" plan based on the literal structure of the question: [1. Identify the author of 'One Hundred Years of Solitude' (Gabriel García Márquez) -> 2. Retrieve all his friends -> 3. Filter for poets from the list of friends -> 4. Verify which poet won a Nobel Prize].

    - However, this plan would likely fail at the second step. García Márquez's social circle was extremely wide, making the task of retrieving "all his friends" nearly impossible and likely to return a massive amount of irrelevant information, rendering subsequent steps ineffective.

    - A monolithic LLM with dynamic correction capabilities, upon observing that the result set from the second step is too large, could immediately abandon the original path and adopt a more optimal backward strategy.
        Does FutureMind risk losing this flexibility to optimize strategy mid-reasoning due to this characteristic?

**Questions:**

see weakness

---

> ### Author Response · Authors · 2025-11-20
> **1. Discussion on Quantifying "Cognitive Compatibility".**
>
> &emsp; We fully agree that introducing quantitative metrics to measure "cognitive compatibility" would strengthen the persuasiveness of our conclusions. In this work, we primarily aim to highlight the phenomenon and provide preliminary explanations. However, in our future work, we have already planned and begun exploring several more direct measures, including (but not limited to):
> - Entropy and distributional differences: average token-level entropy, JS/KL divergence between teacher and student next-token distributions;
> - Strategy representation similarity: similarity based on hidden representations or strategy-specific features.

---

> ### Author Response · Authors · 2025-11-20
> **2. Applicability of FutureMind in Open-Domain Question Answering.**
>
> &emsp; It should be emphasized that the core of the proposed FutureMind modular reasoning framework lies in its structured **P-L-S-R** problem-solving workflow. We argue that any type of problem — whether closed-domain or open-domain — can be formulated in a structured manner through problem decomposition and identification of critical conditions. The key distinction is that, in closed-domain tasks, critical conditions typically appear as explicit and verifiable hard constraints, whereas in open-domain or creative tasks, they often manifest as implicit and flexible soft constraints.
>
> &emsp; Based on this perspective, we contend that FutureMind is highly transferable to open-domain and creative tasks, requiring only domain-specific adaptation during the strategy planning stage.
>
> &emsp; To illustrate this, we take code generation as an example. During the problem analysis stage, FutureMind structures user requirements into functional units, constraints, and acceptance criteria, including functional decomposition, input/output specification, and identification of critical boundary conditions. During the logical reasoning stage, the framework shifts from deterministic causal-chain reasoning to a form of "soft-logic planning" suitable for code generation, emphasizing structural factors such as module dependencies, priorities, and interface consistency.
>
> &emsp; On this basis, code-generation-oriented strategy guidance can be constructed through, for example:
> - Iterative Generation Strategy that refine and revise over multiple rounds;
> - Backward Generation Strategy that unfolds from target outputs.
>
> &emsp; It should be noted that our evaluation of FutureMind’s potential in code generation is limited to preliminary internal validation on the problem analysis and logical reasoning stages. Systematic abstraction and summarization of strategy-guidance methods for diverse open-domain tasks remain ongoing work.

---

> ### Author Response · Authors · 2025-11-20
> **3. Exploration of FutureMind’s Flexibility in Correcting the Logical Path.**
>
> &emsp; We thank the reviewer for raising this question and providing a concrete example. This scenario is precisely what FutureMind is designed to address. Rather than functioning as a one-shot, fixed-plan generator, FutureMind is designed as a **dynamically callable tool** (clarified in Appendix~E.3) that can be invoked iteratively as needed. This design preserves structured planning capabilities while maintaining flexibility in reasoning, thereby enabling more adaptive inference strategies. Its core mechanisms include:
>
> - On-demand triggering. The student model can invoke FutureMind whenever it encounters an execution bottleneck at any reasoning step. Examples include candidate sets being too large, retrieval noise being too high, or intermediate results becoming invalid. In such cases, the model retrieves alternative or refined strategy suggestions.
>
> - Strategy diversity with decentralized control. FutureMind provides dynamic and diverse reasoning strategies, while the ultimate selection and adjustment of the reasoning path are made by the student model based on real-time retrieval results and available computational budget, preserving flexibility.
>
> &emsp; Thus, when facing an overly broad retrieval issue, the student model dynamically invokes FutureMind to form a closed loop of **execution bottleneck → on-demand invocation → strategy optimization**, mirroring the iterative problem-solving logic of human experts.
>
> &emsp; To illustrate this loop more concretely, we provide a typical execution trajectory using the reviewer’s example:
>
> &emsp; **Problem:** Who is the poet that was a friend of the author of "One Hundred Years of Solitude" and also won a Nobel Prize?
>
> &emsp; **First FutureMind call**. During the initial exploration, the model recognized that retrieving "friends" could lead to a large number of redundant candidates. It therefore made the first FutureMind call and obtained a **forward stepwise reasoning strategy**: Identify the author → Retrieve well-known literary associates → Filter for poets → Verify Nobel laureateship.
>
> &emsp; **Second FutureMind call**. During execution, even after narrowing the set of "associates", dozens of candidates remained, indicating high downstream reasoning difficulty. The model triggered a second FutureMind call and received a recommended **backward constraint–focusing strategy**: List Nobel Prize–winning poets → Filter candidates with documented associations to García Márquez → Confirm friendship via credible sources.
>
> &emsp; After receiving this guidance, the model refined the second step to "filter candidates with publicly verifiable interaction records with García Márquez" and proceeded. It first identified **Pablo~Neruda** (1971 Nobel Prize in Literature, poet) as the core candidate and confirmed their long-term friendship via Márquez’s essay **García Márquez on Pablo Neruda**. Verification through literary archives ultimately yielded the correct answer **Neruda**.

---

> > ### Comment · Reviewer_4F63 · 2025-11-21
> >
> > Your clarification aligns closely with my understanding. Under the current circumstances, I believe maintaining the original positive rating represents the most objective and appropriate scoring decision. That said, I have increased the score for soundness, and I hope you will incorporate our discussion into the new version of the manuscript.

---

> > > ### Author Response · Authors · 2025-11-21
> > >
> > > Thank you for your kind feedback and for increasing the score for soundness, we will incorporate our discussion into the revised manuscript accordingly.

---

### Author Response · Authors · 2025-11-21
**Appreciation and Overall Response.**

&emsp; We sincerely thank all reviewers for their time, constructive feedback, and insightful suggestions. We have carefully addressed every comment and revised the manuscript accordingly. **A improved version of the paper has now been uploaded.** We kindly invite the reviewers to read our responses in conjunction with the revised manuscript, as this will provide clearer context and help illustrate how each concern has been resolved.

&emsp; We deeply appreciate the reviewers’ efforts in helping us strengthen the clarity, technical soundness, and we look forward to further feedback from the reviewers.

---

### Author Response · Authors · 2025-12-02
**Summary Comment**

&emsp; **We sincerely thank all reviewers for their constructive feedback and valuable suggestions.**

&emsp; We propose **FutureMind**, named to reflect our vision for future AI systems capable of flexibly applying structured reasoning strategies, similar to human experts. Our goal is to guide models to understand problem structure, plan retrieval strategies adaptively, and reduce unnecessary retrieval calls, achieving more efficient and robust inference. The framework allows a smaller student model to selectively leverage thinking-pattern priors from a larger teacher model **without any additional training**, offering a new paradigm for structured, expert-like reasoning in challenging tasks.

&emsp; Among the three reviews, two reviewers (scores: 6, 6) mainly provided suggestions and discussions regarding implementation clarity, completeness, and open-ended questions. One reviewer (score: 4) raised concerns regarding the definitions of concepts such as "knowledge distillation" and "cognitive bias bottleneck", as well as the abstraction level of Section 3.

&emsp; **Core Concept Clarifications**
- **Reasoning Distillation.** FutureMind aligns with the reasoning-distillation paradigm by providing strategy-level priors from the teacher model, rather than transferring traditional soft labels. This is clearly stated in the first edition of the paper.
- **Teacher Role.** The teacher model provides a high-level framework of logical reasoning and principles for tool usage, while the student retains full execution control, ensuring flexibility and adaptive reasoning.
- **Cognitive Bias Bottleneck & Strategic Information Loss.** The observed non-monotonic performance patterns indicate that a smaller cognitive gap between teacher and student facilitates more effective strategy uptake. Strategic information loss refers to the simplification loss of essential strategic knowledge that occurs when the teacher’s reasoning becomes too complex or too far beyond what the student can internalize.

&emsp; **Method Applicability and Flexibility**
- Although our experiments focus on knowledge-intensive settings, FutureMind is **broadly applicable to open-domain and creative tasks** via problem decomposition, identification of critical conditions, and soft-logic planning.
- Its **iterative, on-demand invocation mechanism** enables dynamic strategy revision during execution, mirroring human expert reasoning processes and addressing concerns about premature commitment to an initial plan.

&emsp; We appreciate that reviewers recognized the clarity and validity of our rebuttal explanations. One reviewer **increased their soundness score** after the rebuttal, suggesting that our clarifications on conceptual definitions, methodology, and broader applicability were helpful and persuasive.

&emsp; We have incorporated all constructive suggestions and clarifications into the revised manuscript, including additional ablation studies, more detailed execution examples for Section 3, expanded computational cost analyses, and other improvements.

&emsp; We hope the AC considers **FutureMind’s vision, contributions, conceptual clarifications, and the positive reviewer responses** when making their decision. **We sincerely thank the AC for their time and thoughtful evaluation.**

---

### Meta-Review · Area_Chair_GfA2 · 2026-01-07

**Summary:**

The authors propose FutureMind, a modular reasoning framework that enhances small language models with distilled strategic thinking and retrieval-guided reasoning from large language models, enabling state-of-the-art multi-hop question answering under zero-training conditions. From my reading of this paper and the review-rebuttal discussion, I think this paper has made essential contributions on the development of SLMs with high quality of writing. Thus I recommend acceptance of this paper.

**Reviewer Concerns:**

I think the reviewers are good in the review and dicussion.

**Reviewer Scores:**

The rebuttal discussions are good for this paper.

---

### Decision · Program_Chairs · 2026-01-26

Accept (Poster)